# Gradients of glucose metabolism regulate morphogen signalling required for specifying tonotopic organisation in the chicken cochlea

James DB O'Sullivan[1], Thomas S Blacker[2], Claire Scott[1], Weise Chang[3], Mohi Ahmed[1], Val Yianni[1], Zoe F Mann[1]*

[1]Centre for Craniofacial and Regenerative Biology, Faculty of Dentistry Oral and Craniofacial Sciences, King's College London, London, United Kingdom; [2]Research Department of Structural and Molecular Biology, University College London, London, United Kingdom; [3]National Institute on Deafness and Other Communication Disorders, National Institutes of Health, Bethesda, United States

**Abstract** In vertebrates with elongated auditory organs, mechanosensory hair cells (HCs) are organised such that complex sounds are broken down into their component frequencies along a proximal-to-distal long (tonotopic) axis. Acquisition of unique morphologies at the appropriate position along the chick cochlea, the basilar papilla, requires that nascent HCs determine their tonotopic positions during development. The complex signalling within the auditory organ between a developing HC and its local niche along the cochlea is poorly understood. Using a combination of live imaging and NAD(P)H fluorescence lifetime imaging microscopy, we reveal that there is a gradient in the cellular balance between glycolysis and the pentose phosphate pathway in developing HCs along the tonotopic axis. Perturbing this balance by inhibiting different branches of cytosolic glucose catabolism disrupts developmental morphogen signalling and abolishes the normal tonotopic gradient in HC morphology. These findings highlight a causal link between graded morphogen signalling and metabolic reprogramming in specifying the tonotopic identity of developing HCs.

*For correspondence: zoe.mann@kcl.ac.uk

**Competing interest:** The authors declare that no competing interests exist.

## Editor's evaluation

Morphogens such as Sonic hedgehog and Bone morphogenetic proteins (BMP) are known to establish the tonotopic organization of the cochlea. In this paper, the authors demonstrated for the first time that differential glucose metabolism along the basilar papilla (chicken cochlea) regulates the gradient of BMP7 signaling required for establishing tonotopy of the cochlea.

## Introduction

Hearing relies upon the life-long function of mechanosensory hair cells (HCs) and their associated glial-like supporting cells (SCs) within the cochlea. In both mammals and birds, different frequencies stimulate HCs located at different positions along the basal-to-apical long axis of the auditory epithelium to separate complex sounds into their spectral components. This phenomenon, known as tonotopy, underlies our ability to differentiate between the high pitch of a mosquito and the low rumbling of thunder. The specific factors regulating the development of tonotopy remain largely unclear. As high-frequency HCs show increased vulnerability to insults, including ageing (*Gordon-Salant, 2005*), noise damage (*Fettiplace and Nam, 2019*; *Wu et al., 2020*), and ototoxicity (*Forge and Richardson,*

**Figure 1.** Pathways of glucose catabolism regulating cellular NADPH/NADH. (pink mitochondrial OXPHOS) – Glucose metabolism in *mitochondria*. Following its conversion from glucose during glycolysis, pyruvate is transported into the mitochondria via the mitochondrial pyruvate carrier (MPC) and enters the tricarboxylic acid (TCA) cycle. Its sequential oxidation provides reducing equivalents in the form of NADH to the electron transport chain (ETC), driving ATP production by oxidative phosphorylation (OXPHOS). (yellow glycolysis) – *Cytosolic* glucose flux via the main branch of glycolysis. In this process, one molecule of glucose is anaerobically converted into two molecules of pyruvate to yield two molecules of ATP. Lactate dehydrogenase (LDH) acts to maintain the pool of NAD⁺ necessary for glycolysis to take place by oxidising NADH upon the reduction of pyruvate to lactate. (green pentose phosphate pathway) – *Cytosolic* glucose flux into the oxidative branch of the pentose phosphate pathway (PPP). Running parallel to glycolysis, the PPP branches off at glucose 6-phosphate (G6P) generating NADPH and ribose 5-phosphate (R5P). PPP shuttles carbons back into the main glycolytic pathway at glyceraldehyde 3-phosphate and fructose 1,6-bisphosphate. Different pathways of glucose flux can be targeted for pharmacological intervention. Inhibitors for various metabolic branch points are indicated in red (UK5099, YZ9, 2-DOG, 6-AN, Shikonin).

*1993*), awareness of the mechanisms underlying the formation of frequency-specific HC properties is crucial to understanding both acquired auditory defects, HC repair and regeneration. Enhanced knowledge of the pathways that drive specification of HC phenotypes at different frequency positions could identify novel strategies to preserve and restore high-frequency hearing loss.

Metabolism, encompassing the complex network of chemical reactions that sustain life (summarised in *Figure 1*), has emerged as a key regulator of cell fate and differentiation (*Ito and Ito, 2016*). Reciprocity between metabolic networks and the epigenome has been extensively studied in models of cancer cell biology and tumourigenesis (*Kinnaird et al., 2016*). Here, chromatin modifying enzymes (involved in histone acetylation and methylation) that drive cell fate switches rely upon metabolic intermediates as cofactors or substrates, highlighting a link between cell metabolism and transcriptional regulation (*Campbell and Wellen, 2018*). Reprogramming between glycolytic and oxidative pathways has also been reported in developing tissues, including migratory neural crest cells (*Bhattacharya et al., 2020*), the zebrafish otic vesicle (*Kantarci et al., 2020*), trophectoderm in the mouse embryo (*Chi et al., 2020*), and the presomitic mesoderm (*Oginuma et al., 2017*; *Oginuma et al., 2020*; *Bulusu et al., 2017*; *Miyazawa et al., 2022*). Nevertheless, a regulatory role for metabolism has not been explored in the context of cell fate and patterning in the developing inner ear epithelia. This is, in part, because the classic biochemical approaches from which our knowledge of metabolism has formed involve the destructive extraction of metabolites from a sample. Probing metabolism in this manner, although valuable, means that any spatial organisation of metabolic pathways in complex tissues is lost. As the cochlea contains multiple cell types, investigating the role of metabolism in the regulation of their development requires experimental approaches capable of interrogating metabolic pathways in live preparations with single-cell resolution.

We have previously demonstrated that fluorescence lifetime imaging microscopy (FLIM) provides a label-free method to identify metabolic differences between inner ear cell types (*Blacker et al., 2014*). By spatially resolving differences in the fluorescence decay of the reduced redox cofactors

nicotinamide adenine dinucleotide (NADH) and its phosphorylated analogue NADPH (*Figure 1*) we can extract information about the metabolic state of a cell (*Blacker et al., 2014*). Here, we apply this technique to investigate a role for metabolism in specifying morphological properties of proximal (high-frequency) verses distal (low-frequency) HCs in the chick cochlea. The HC phenotypes associated with different tonotopic positions are defined using previously characterised morphometrics as read-outs (*Tilney et al., 1992*; *Goodyear and Richardson, 1997*). By applying NAD(P)H FLIM in different regions of the developing basilar papilla (BP), we identify a gradient in NADPH-linked glucose metabolism along the tonotopic axis. The NAD(P)H gradient did not originate from a tonotopic switch between glycolytic and oxidative pathways or from differences in cellular glucose uptake. We find that the metabolic gradient along the developing BP originates instead from tonotopic differences in the catabolic fate of cytosolic glucose once it has entered the cell. By modulating the flux of glucose through specific cytosolic branches, we systematically interrogated its role in specifying tonotopic properties of developing HCs. We found that the cellular balance of glucose entering the pentose phosphate pathway (PPP) and the main branch of glycolysis (*Figure 1*) instructs tonotopic HC morphology by regulating the graded expression of Bone morphogenetic protein 7 (Bmp7) and its antagonist Chordin-like-1 (Chdl1), known determinants of tonotopic identity (*Mann et al., 2014*). This work highlights a novel role for cytosolic glucose metabolism in specifying HC positional identity at the morphological level providing the first evidence of a link between metabolism and morphogen signalling in the developing inner ear.

## Results

### NAD(P)H FLIM reveals differences in the cellular balance between NADPH and NADH along the tonotopic axis of the developing BP

NAD and NADP are metabolic cofactors responsible for ferrying reducing equivalents between intracellular redox reactions throughout the cellular metabolic network (*Figure 1*). The two molecules are fluorescent in their reduced (electron-carrying) forms NADH and NADPH, a feature that is lost upon oxidation to $NAD^+$ and $NADP^+$. The spectral properties of NADH and NADPH are identical, meaning that their combined signal emitted from living cells is labelled as NAD(P)H. FLIM of NAD(P)H has shown significant promise for identifying changes in the metabolic pathways active at a given location in living cells.

NAD(P)H FLIM typically resolves two fluorescence lifetime components in live cells and tissues one with a duration of around 0.4 ns coming from freely diffusing NAD(P)H ($\tau_{free}$) and the other of 2 ns or more from the pool of NAD(P)H that is bound to enzymes and cofactors ($\tau_{bound}$) (*Skala et al., 2007*; *Yu and Heikal, 2009*; *Figure 2B*). Changes in the duration of $\tau_{bound}$ indicate switching in the enzyme families that are bound to the overall NAD(P)H population. NAD(P)H FLIM can therefore report changes in metabolic state in live cells during physiological processes. We used NAD(P)H FLIM to monitor metabolism along the proximal-to-distal (tonotopic) axis of the BP during development (*Figure 2E–H*). The gradient observed in $\tau_{bound}$ throughout BP development, specifically at E6 and E9, is consistent with our previous work, where we showed that graded morphogen signalling along the BP establishes HC positional identity between E6 and E8 (*Mann et al., 2014*; *Thiede et al., 2014*). Around E6, when cells in the BP begin acquiring their positional identity (*Mann et al., 2014*; *Thiede et al., 2014*), we observed a significant difference in $\tau_{bound}$ along the tonotopic axis (*Figure 2D–H*). The proximal-to-distal gradient in $\tau_{bound}$ (*Figure 2D*) was also evident at E9, when a majority of cells are post mitotic (*Katayama and Corwin, 1989*), and at E14, when HCs functionally resemble those in a mature BP (*Figure 2E, F*). Changes in $\tau_{bound}$ report shifts in the cellular balance between NADPH and NADH (*Blacker et al., 2014*; *Blacker et al., 2013*; *Gafni and Brand, 1976*), which reflects glucose catabolism in distinct branches of glycolysis (*Figure 2H*). These data therefore suggest alterations in the balance between NADH- and NADPH-linked glucose metabolism along the tonotopic axis of the developing BP (*Figure 2H*).

### Live imaging of mitochondrial metabolism and glucose uptake along the tonotopic axis of the developing BP

The redox states of the cellular NAD(P) pools are highly interrelated with the balance between cytosolic glycolysis and mitochondrial oxidative phosphorylation (OXPHOS) (*Russell et al., 2022*; *Ying, 2008*).

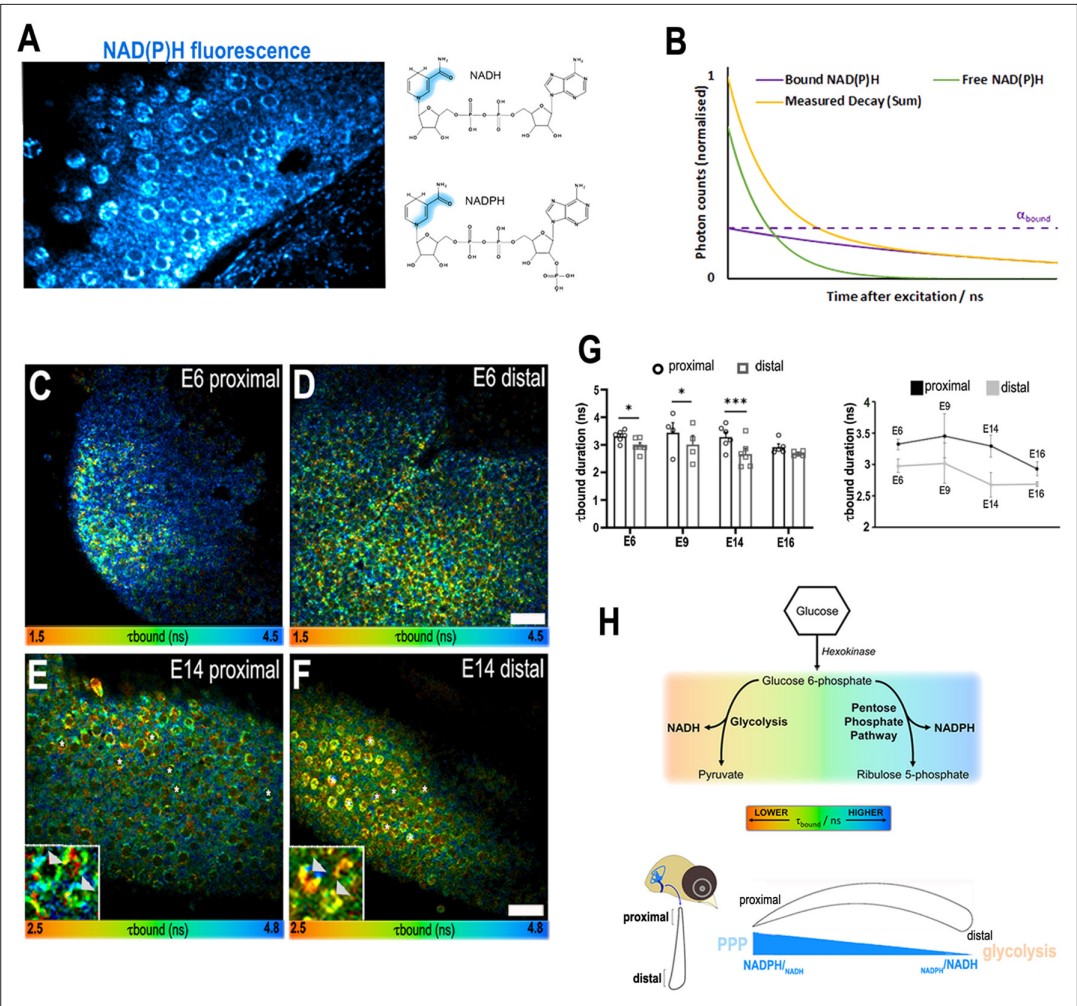

**Figure 2.** A proximal-to-distal metabolic gradient in the developing chick cochlea. (**A**) Two-photon fluorescence image showing NAD(P)H in a live basilar papilla (BP) explant at E14 and the origin of inherent fluorescence at the nicotinamide ring. (**B**) NAD(P)H fluorescence lifetime imaging microscopy (FLIM) resolves two components corresponding to freely diffusing (shorter lifetime, $\tau_{\text{free}}$) and enzyme bound (longer lifetime, $\tau_{\text{bound}}$). Changes in $\tau_{\text{bound}}$ imply changes in the specific enzymes to which NAD(P)H is binding and, therefore, the metabolic state of the cell. The proportion of the total NAD(P)H population that is bound to enzymes, labelled $\alpha_{\text{bound}}$, determines the relative contribution of the two species immediately after excitation. (**C–F**) FLIM images of the bound NAD(P)H fluorescence lifetime signal $\tau_{\text{bound}}$ in the proximal and distal BP regions at E6 and E14. White asterisks indicate the hair cells (HCs). Higher magnification images highlight the differences in $\tau_{\text{bound}}$ between proximal and distal HCs at E14 (arrowheads). (**G**) Quantification of $\tau_{\text{bound}}$ during development shows a shift from NADPH to NADH producing pathways. Line graphs highlight differences in $\tau_{\text{bound}}$ between proximal (black) and distal (grey) BP regions throughout development. Scale bars = 50 µm. Data are mean ± standard error of the mean (SEM); E6: $n$ = 6, E9: $n$ = 4, E14: $n$ = 6, and E16: $n$ = 5 biological replicates. *p < 0.05, ***p < 0.001 two-way analysis of variance (ANOVA). (**H**) Schematic of the chick BP, indicating the proximal and distal regions. Proposed gradient in cellular NADPH/NADH and thus glucose flux along the developing BP. Bottom schematic depicts interpretation of the $\tau_{\text{bound}}$ lifetime signal reported by NAD(P)H FLIM along the proximal-to-distal axis. The gradient in $\tau_{\text{bound}}$ duration reflects differences in fate of glucose catabolism. Short lifetimes (orange) indicate NADH production and therefore glucose flux through the main glycolytic pathway. Longer lifetimes (blue) indicate NADPH production and glucose catabolism in the pentose phosphate pathway (PPP). Differences in the $\tau_{\text{bound}}$ lifetime duration thereby confer differences in the catabolic fate of glucose.

We therefore tested whether the gradient in cellular NADPH/NADH reflected a progressive shift from glycolytic to mitochondrial OXPHOS, differences in cellular glucose uptake, or in the catabolic fate of glucose by conducting live imaging in BP explants using a range of metabolic indicators. To assess mitochondrial metabolism, we used the potentiometric fluorescent probe tetramethyl-rhodamine-methyl-ester (TMRM), a cell permeable dye that reports mitochondrial membrane potential ($\Delta \Psi_{mt}$) in living cells (*Duchen et al., 2003*). TMRM reports glycolytically derived pyruvate oxidation in the mitochondrial tricarboxylic acid (TCA) cycle and the activity of the mitochondrial electron transport chain (*Figure 1*). 2-(N-(7-Nitrobenz-2-oxa-1,3-diazol-4-yl)Amino)-2-Deoxyglucose (2-NBDG) is a fluorescent glucose analogue that when transported into cells via glucose (GLUT) transporters provides an estimate of cellular glucose uptake (*Yamada et al., 2007*). Thus, the 2-NBDG fluorescence measured in

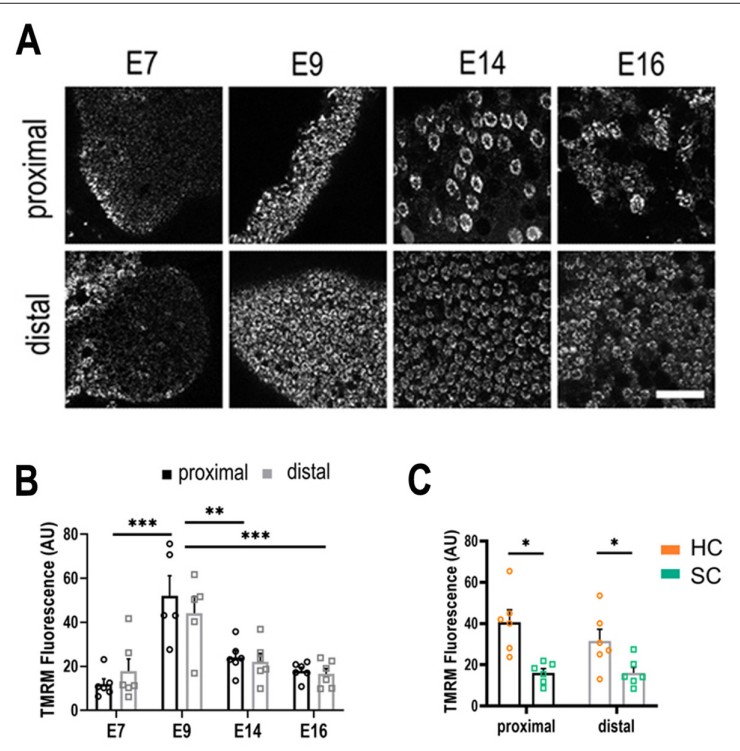

**Figure 3.** Live imaging of mitochondrial metabolism in hair cells (HCs) and supporting cells (SCs) at different positions along the tonotopic axis. (**A**) Mitochondrial membrane potential measured using tetramethyl-rhodamine-methyl-ester (TMRM) in single z-planes from image stacks in the proximal and distal regions of live basilar papilla (BP) explants. (**B**) TMRM fluorescence indicates a significant increase in mitochondrial activity between E7 and E9, followed by significant decrease between E9 and E14. (**C**) Differences in mitochondrial activity (TMRM fluorescence) between HCs and SCs along the tonotopic axis at E14. Data are mean ± standard error of the mean (SEM). **p > 0.01, ***p < 0.001 for proximal and distal regions two-way analysis of variance (ANOVA). E7: $n = 6$, E9: $n = 5$, E14: $n = 6$, E16: $n = 6$ biological replicates. HCs versus SCs $n = 6$ biological replicates *p > 0.05 two-way ANOVA. Scale bar = 40 µm.

The online version of this article includes the following source data and figure supplement(s) for figure 3:

**Figure supplement 1.** Tetramethyl-rhodamine-methyl-ester (TMRM) fluorescence intensity is not dependent on uptake via the hair cell (HC) transduction channel.

**Figure supplement 2.** Live imaging of glucose uptake in hair cells (HCs) and supporting cells (SCs) at different positions along the tonotopic axis.

**Figure supplement 3.** Differential expression of metabolic enzymes along the tonotopic axis of the chick basilar papilla (BP).

**Figure supplement 3—source data 1.** This source data contains the vaules for affymetrix and bulk RNA-seq analysis at E6.5 and E14.

**Figure supplement 4.** Tonotopic expression of *Bmp7* along the developing basilar papilla (BP).

**Figure supplement 5.** Negative controls for RNA scope analysis.

a given cell after a defined period of loading reflects the rate of glucose uptake by that cell (*Yamada et al., 2007*).

Explants were dual loaded with 350 nM TMRM and 1 mM 2-NBDG and both fluorescence signals were analysed from single cells between E7 and E16. TMRM fluorescence revealed no significant difference in $\Delta\Psi_{mt}$ between proximal and distal regions at any developmental stage (*Figure 3A, B*). Analysis of TMRM fluorescence revealed a consistently higher $\Delta\Psi_{mt}$ in fully differentiated HCs compared to SCs from E14 onwards (*Figure 3C*). To rule out whether the higher TMRM fluorescence occurred due to differences in dye uptake via the HC transduction channel, explants were dual loaded with the permeant mechanoelectrical transduction (MET) channel blocker FM1-43 (*Gale et al., 2001*; *Figure 3—figure supplement 1*). These findings indicate no significant difference in mitochondrial activity along the tonotopic axis throughout development, suggesting that the gradient in NADPH/NADH reported by $\tau_{bound}$ (*Figure 2C–G*) does not arise from variations in the balance between cytosolic and mitochondrial ATP production, as often observed in development (*Bhattacharya et al., 2021*) but from differences in the specific route utilised for the processing of glucose in the cytosol. This was supported by measurements of 2-NDBG fluorescence in the same cells (*Figure 3—figure supplement 2A*). These analyses revealed no differences in glucose uptake along the tonotopic axis at any developmental stage or between cell types (*Figure 3—figure supplement 2B, C*), suggesting differences in the fate rather than overall flux of glucose underpin the gradient in $\tau_{bound}$ (i.e., NADPH/NADH).

## Tonotopic expression of metabolic mRNAs along the proximal-to-distal axis of the developing BP

To further probe the biochemical basis for the gradient in $\tau_{bound}$ we exploited existing transcriptional data sets generated from proximal and distal regions of the developing BP (*Mann et al., 2014*). Prior to mRNA isolation for bulk RNA-seq and Affymetrix microarray analysis, BPs were separated into proximal, middle, and distal thirds. Data were then analysed for differential expression of metabolic mRNAs involved in NADPH regulation and cytosolic glucose flux at E6.5 and E14 (*Figure 3—figure supplement 3A*; *Mann et al., 2014*). From the combined data sets, we identified multiple genes involved in NADPH-linked glucose metabolism with differential expression along the tonotopic axis (*Figure 3—figure supplement 3A*). Expression of these genes was verified using RNA scope and Immunohistochemistry. The only enzyme linked with cytosolic glucose flux and cellular NADPH/NADH showing a consistent differential expression throughout development, using all three validation methods, was pyruvate kinase M2 (Pkm2) (*Figure 3—figure supplement 3A*, *Figure 4—figure supplements 1–3*). No probe controls were used to validate the labelling (*Figure 3—figure supplement 5*).

## Pkm2 protein is expressed tonotopically along the developing BP

Pyruvate kinase M2 (Pkm2) is a unique splice isoform of the enzyme pyruvate kinase (PK) and catalyses the final rate-limiting step in glycolysis. Pkm2 regulates the activity of metabolic enzymes in the upper branch of glycolysis by acting as a gatekeeper and diverting glucose flux towards pyruvate production or into the PPP (*Grüning et al., 2011*). Given this regulatory role in the catabolic fate of cytosolic glucose (*Grüning et al., 2011*), and the fact that increased Pkm2 activity is linked with higher cellular NADPH (*Yang and Lu, 2015*), we hypothesised that in correlation with the $\tau_{bound}$ gradient, Pkm2 expression would be higher in the proximal region. Consistent with the observed gradient in NAD(P)H (*Figure 2*) and in Pkm2 mRNA levels (*Figure 3—figure supplement 3A*, *Figure 4—figure supplements 1–3*), we show higher Pkm2 protein expression in HCs but not SCs at the proximal compared to distal end of the BP between E9 and E14 (*Figure 4*).

## Higher intracellular pH in the proximal BP favours Pkm2 activity associated with an increased NADPH/NADH ratio and glucose flux into the PPP

The metabolic function of Pkm2 is determined by whether the enzyme exists as a tetramer rather than a dimer. In its dimeric form, Pkm2 functions as a metabolic switch, diverting glucose towards the PPP for biosynthesis or towards pyruvate for energy production (*Nandi et al., 2020*). Allosteric modifications regulating the ratio between the tetrameric and dimeric forms of Pkm2 are driven by factors in the surrounding environment including intermediate metabolites and pH (*Nandi et al.,*

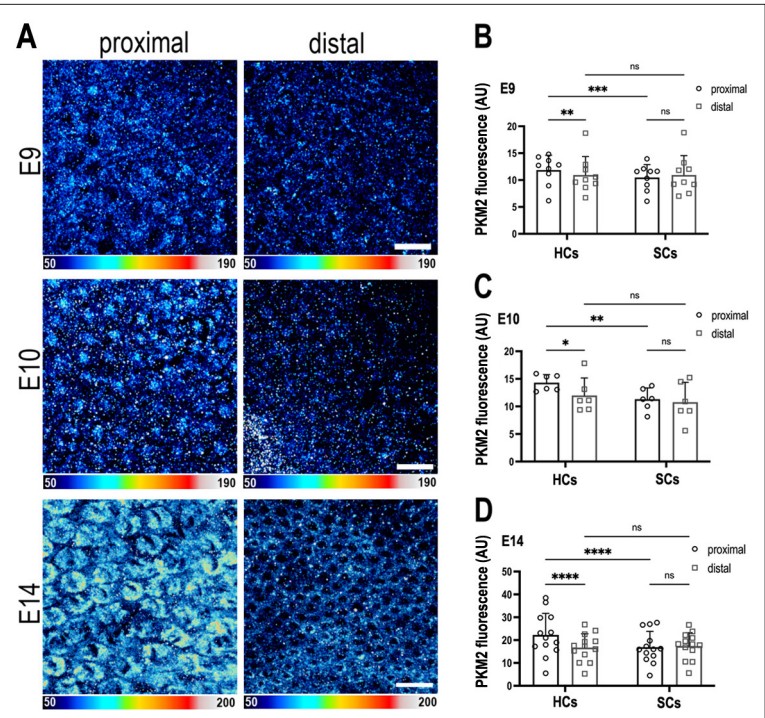

**Figure 4.** The metabolic gatekeeper Pkm2 is expressed in a tonotopic gradient during basilar papilla (BP) development. (**A**) BP whole-mounts labelled for the metabolic enzyme Pkm2 in proximal and distal regions throughout development (E9, E10, and E14). Images show Pkm2 expression at the level of the hair cell (HC) nuclei. Epithelial z-position was determined using Phalloidin and Calbindin staining within the same preparation (images not shown). (**B–D**) Quantification of Pkm2 fluorescence intensity in proximal and distal BP regions at E9, E10, and E14. HC and supporting cell (SC) regions of interest (ROIs) were determined using Phalloidin and Calbindin staining within the same preparation. Data are mean ± standard error of the mean (SEM). E9: $n = 9$, E10: $n = 6$, E14: $n = 13$ independent biological replicates. *$p = <0.05$, **$p = <0.01$, ***$p = <0.001$, ****$p = <0.0001$ two-way analysis of variance (ANOVA). Scale bars are 20 μm.

The online version of this article includes the following figure supplement(s) for figure 4:

**Figure supplement 1.** Proximal-to-distal expression of *Sox2* and *Pkm2* at E8.

**Figure supplement 2.** Tonotopic expression of *Pkm2* in the developing basilar papilla (BP).

**Figure supplement 3.** Tonotopic expression of *Pkm2* and *Atoh1* in the developing basilar papilla (BP).

---

*2020*; *Zhang et al., 2019*). Given the pH-dependent nature of PKM2 allostery and that the main rate-limiting enzymes driving PPP-linked glucose metabolism display optimal activity at alkaline cytosolic pH (*Alfarouk et al., 2020*), we next investigated differences in intracellular pH (pH$_i$) along the tonotopic axis using the indicator pHrodo Red. When using this probe, low pHrodo Red fluorescence reflects an alkaline pH and high fluorescence a more acidic pH.

Explants were dual loaded with the pH$_i$ indicator pHrodo Red (*Figure 5*) and the live probe SIR-actin to distinguish HCs from SCs (*Figure 5—figure supplements 1 and 2*). We identified opposing proximal-to-distal gradients in pH$_i$ in HCs and SCs along the tonotopic axis, using pHrodo Red, which reported a more alkaline pH$_i$ in HCs at the proximal compared to distal end of the organ (*Figure 5*). The higher pH$_i$ in the proximal region reflects a metabolic phenotype consistent with higher PPP activity and dimeric Pkm2. Overall, the higher pH and Pkm2 expression levels and the possible dimeric confirmation are consistent metabolically with a longer $\tau_{bound}$ (NAD(P)H lifetime). To investigate whether the proximal-to-distal gradient in pH was maintained at later developmental stages, we also quantified the pHrodo Red signal in HCs and SCs at E14. At later developmental stages, we find the pH gradients to be reversed (*Figure 5—figure supplement 3*). As tonotopic patterning and positional identity are specified between E6 and E7.5 (*Mann et al., 2014*), the gradient at E14 is unlikely to impact the gradient in HC morphology.

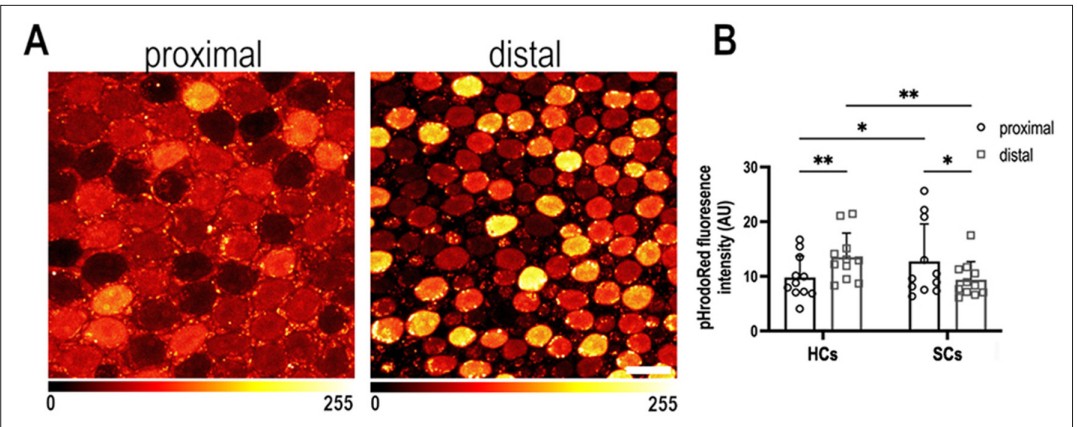

**Figure 5.** Intracellula pH varies as a function of frequency position during early basilar papilla (BP) development. (**A**) Intracellular pH, reported by pHrodo Red fluorescence intensity, in the proximal and distal BP regions at E9. High fluorescence indicates acidic pH and low fluorescence a more alkaline pH. (**B**) Mean pHrodo Red fluorescence in measured from hair cells (HCs) and supporting cells (SCs) in proximal and distal frequency BP regions. Note the proximal-to-distal gradient in intracellular pH. Data are mean ± standard error of the mean (SEM) from 11 independent biological replicates for HCs and 12 independent biological replicates for SCs. *p = <0.05, **p = <0.01 two-way analysis of variance (ANOVA). Scale bar is 10 μm.

The online version of this article includes the following figure supplement(s) for figure 5:

**Figure supplement 1.** Generating hair cell (HC) and supporting cell (SC) analysis masks.

**Figure supplement 2.** Hair cell (HC) and supporting cell (SC) masks used for quantification of pHrodo Red fluorescence.

**Figure supplement 3.** pHrodo Red fluorescence along the tonotopic axis of the chick basilar papilla (BP) at E14.

## Cytosolic glucose metabolism is necessary for tonotopic patterning in the chick BP

Having identified tonotopic gradients in NAD(P)H $\tau_{bound}$ and Pkm2 expression, we investigated a functional role for metabolism in tonotopic patterning by systematically inhibiting glucose flux into different metabolic pathways (*Figure 1*). First, we blocked the entirety of cytosolic glucose metabolism using 2-deoxy-D-glucose (2-DOG), an inhibitor of the enzyme hexokinase (*Barban and Schulze, 1961*), which occurs upstream of the branching of PPP and glycolysis. BP explants were established at E6.5 and maintained for 7 days in vitro (DIV) to the equivalent of E13.5, in control medium or that containing 2 mM 2-DOG supplemented with 5 mM sodium pyruvate (NaP), ensuring adequate substrate supply to the TCA cycle. In a normal BP, proximal HCs have larger luminal surface areas and cell bodies and are more sparsely populated compared to those in the distal region (*Tilney et al., 1992*; *Goodyear and Richardson, 1997*). These morphological gradients are recapitulated in BP explant cultures during development (*Mann et al., 2014*). Here, these metrics were determined by measuring differences in the HC lumenal surface area, the size of HC nuclei and the HC density within defined regions of interest (ROIs) (100 × 100 μm$^2$) along the length of the organ. Lumenal surface area was measured using Phalloidin staining at the cuticular plate and nuclear size with DAPI (*Figure 6A*).

In control cultures, HCs developed with the normal tonotopic morphologies (lumenal surface area, nuclear size and gross bundle morphology) (*Figure 6A, C, D*, *Figure 6—figure supplement 1*). In contrast, when glucose catabolism was blocked between E6.5 and E13.5 equivalent, tonotopic patterning was abolished. This was indicated by a uniformly more distal-like HC phenotype along the BP (*Figure 6A, B*). In addition to changes in HC morphology, treatment with 2-DOG caused a significant increase in HC density in the proximal but not distal BP region (*Figure 6—figure supplement 2A, C*) again consistent with loss of tonotopic patterning along the organ.

Changes in glucose metabolism have been linked with reduced cellular proliferation (*Zhao et al., 2019*). We therefore investigated the effects of 2-DOG on proliferation in developing BP explants. We hypothesised that because the majority of cells in the BP are postmitotic by E10 (*Katayama and Corwin, 1989*), adding 5-ethynyl-2'-deoxyuridine (EdU) to cultures in the presence and absence of

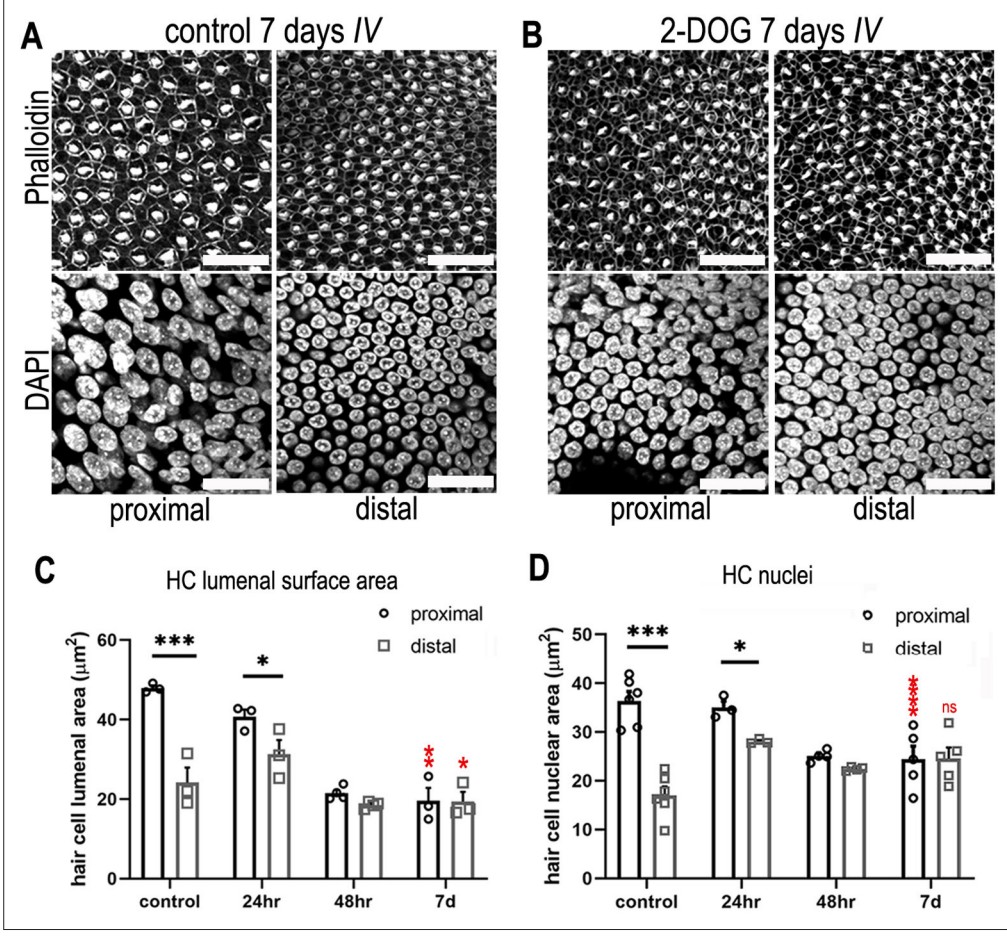

**Figure 6.** Blocking cytosolic glucose metabolism at key stages of cochlear development induces distal-like phenotypes in the proximal basilar papilla (BP). (**A, B**) Maximum z-projections of BP explants showing Phalloidin and DAPI staining in the proximal and distal regions. Explants were maintained from E6.5 for 7 days in vitro (equivalent to E13.5) in either control medium or medium supplemented with 2 mM 2-deoxy-D-glucose (2-DOG) + 5 mM sodium pyruvate (NaP). Phalloidin staining depicts differences in hair cell (HC) morphology between proximal and distal regions and DAPI indicates the gradient in HC size. (**C**) HC lumenal surface area measured in 2500 $\mu m^2$ regions of interest (ROIs) in the proximal (black bars) and distal (grey bars) BP regions for all culture conditions. In controls, mean lumenal surface decreases progressively from the proximal-to-distal region. This gradient is abolished if glucose catabolism is blocked with 2-DOG between E6.5 and E13.5. 2-DOG caused a significant decrease in HC size in the proximal but not distal region. 2-DOG treatments were reduced to 24 or 48 hr to identify the developmental time window during which glycolysis takes effect. Following wash-out of 2-DOG after 24 hr, explants developed with normal HC positional identity. Explants treated with 2-DOG for 48 hr showed no recovery of positional identity following wash-out indicated by the flattening of HC morphology along the BP. (**D**) Quantification of HC nuclei area in the same 2500 $\mu m^2$ ROI areas. Treatment with 2-DOG induced similar, yet less pronounced effects to those seen at the HC cuticular plate. Data are mean ± standard error of the mean (SEM). *$p < 0.05$, ***$p < 0.001$ two-way analysis of variance (ANOVA). Controls; $n = 6$; controls LSA; $n = 3$; 2-DOG, $n = 5$; 24 2-DOG, $n = 3$; and 48 hr 2-DOG, $n = 3$ biological replicates. Red stars indicate two-way ANOVA tests between proximal control and proximal 2-DOG and distal control and distal 2-DOG conditions. To ensure adequate substrate supply to the tricarboxylic acid (TCA) cycle, 2-DOG-treated explants were supplemented with NaP. G6P – glucose 6-phosphate, F6P – fructose 6-phosphate, F16BP – fructose 1,6-bisphosphate, 2-DOG – 2-deoxyglucose. Scale bars are 20 µm.

The online version of this article includes the following figure supplement(s) for figure 6:

**Figure supplement 1.** Morphological differences in the hair cell (HC) bundle and cuticular plate region in basilar papilla (BP) explants treated with control or 2-deoxy-D-glucose (2-DOG) containing medium.

**Figure supplement 2.** 2-Deoxy-D-glucose (2-DOG) increases hair cell (HC) density in the proximal basilar papilla (BP) region independently of proliferation.

*Figure 6 continued on next page*

*Figure 6 continued*

**Figure supplement 3.** Modulating *S*-adenosyl methionine during development abolishes the gradient in hair cell morphology along the tonotopic axis of the basilar papilla (BP).

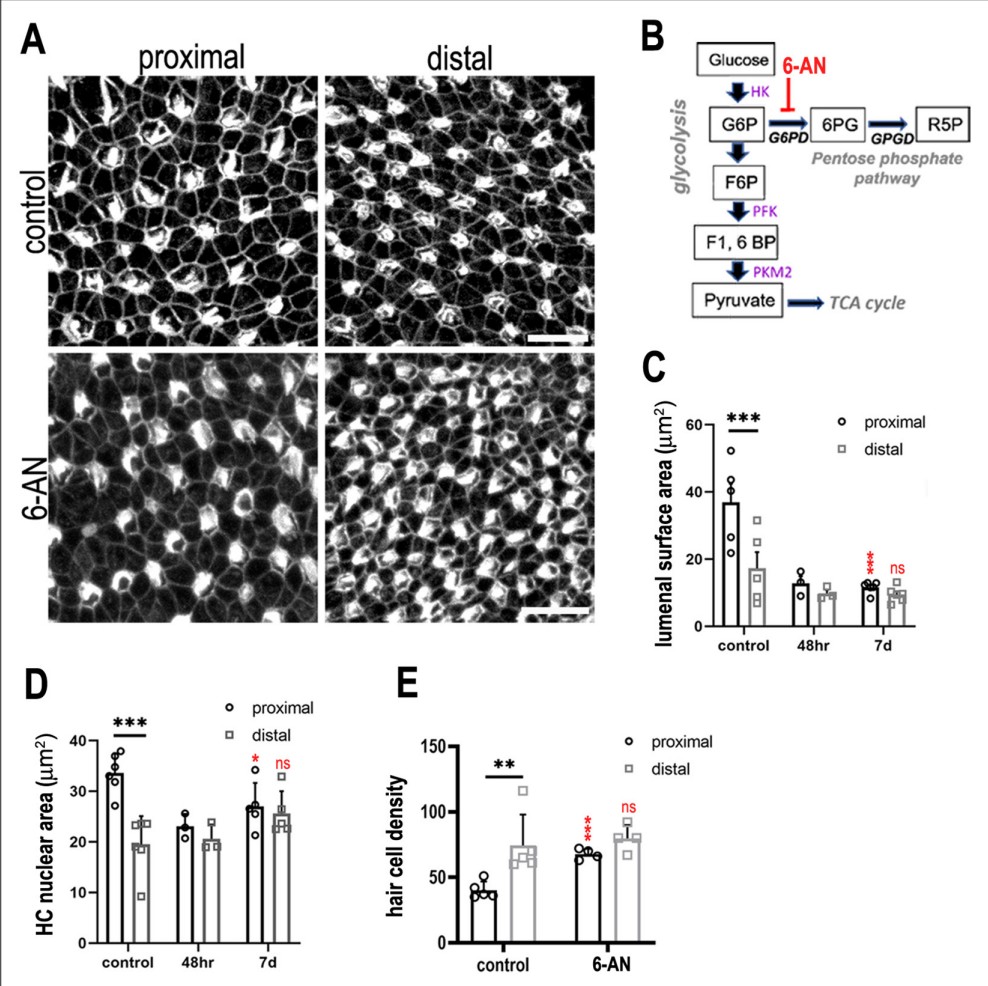

**Figure 7.** Glucose flux through the pentose phosphate pathway modulates hair cell development and positional identity along the tonotopic axis of the basilar papilla (BP). (**A, B**) Maximal z-projections of BP explants cultured from E6.5 for 7 days in vitro with control medium or medium containing 2 µM 6-aminonicotinamide (6-AN). Images show the epithelial surface in proximal and distal BP regions stained with Phalloidin. (**C**) Treatment of explants with 6-AN, a specific blocker of flux through the pentose phosphate pathway, caused a significant reduction in proximal compared to distal hair cell lumenal surface area. (**D, E**) Quantification of hair cell luminal surface area, nuclear area and cell density in defined 2500 µm² regions of interest (ROIs) from the proximal (black bars) and distal (grey bars) BP regions. Data are mean ± standard error of the mean (SEM). **p = <0.01, ***p = <0.001 two-way analysis of variance (ANOVA). Nuclei: controls n = 6, 48 hr n = 3, 7 days n = 5; LAS: controls n = 5, 48 hr n = 3, 7 days n = 5; cell density: controls n = 5, 7 days n = 4. Independent biological replicates. Red stars indicate statistical significance for proximal and distal regions when compared between control and 6-AN conditions. Scale bars are 10 µm. G6P – glucose 6-phosphate, F6P – fructose 6-phosphate, F16BP – fructose 1,6-bisphosphate.

The online version of this article includes the following figure supplement(s) for figure 7:

**Figure supplement 1.** Differences in the hair cell (HC) lumenal surface area and gross bundle morphology in proximal and distal basilar papilla (BP) regions from explants treated with control or 6-aminonicotinamide (6-AN) containing medium.

**Figure supplement 2.** Inhibiting phosphofructokinase alters hair cell (HC) morphology at the luminal surface area during basilar papilla (BP) development.

2-DOG for 48 hr between E8 and E10 would capture any 2-DOG-dependent differences in proliferative capacity. We observed a consistent reduction in proliferation throughout the whole explant when glucose metabolism was blocked with 2-DOG (*Figure 6—figure supplement 2B, D*). Increased proliferation is therefore unlikely to account for the higher cell density observed in the proximal region following the inhibition of glycolysis. Further studies are needed to determine the specific mechanisms underlying this frequency-specific increase in HC density.

As shown in our previous work, reciprocal morphogen gradients of Bmp7 and Chdl1 establish HC positional identity at the morphological level along the developing BP between E6.5 and E8 (*Mann et al., 2014*). To determine whether cytosolic glucose metabolism acts during this same developmental window, we blocked hexokinase activity for defined periods during BP development using 2-DOG. Explants were established at E6.5 and treated for either 24 or 48 hr followed by wash out with control medium.

These treatments correspond to the developmental window (E6.5–E8) described previously for refinement of tonotopic morphologies in developing HCs along the proximal-to-distal axis (*Mann et al., 2014*). The gradient in HC morphology developed normally in BPs treated with 2-DOG for 24 hr but was absent in those treated for 48 hr (*Figure 6C–D*, *Figure 7A & C and D*). These results suggest that glucose metabolism acts within the same developmental time window as Bmp7 and Chdl1 to set up tonotopic patterning along the BP. These findings are also consistent metabolically with the proximal-to-distal gradient in NADPH/NADH ($\tau_{bound}$) observed at E6, E9, and E14 (*Figure 2G*).

To further confirm a role for cytosolic glucose metabolism in establishing HC positional identity, we employed a second method of modulating the pathway independently of hexokinase activity (*Pascale et al., 2019*). The rate of cytosolic glycolysis can be lowered indirectly by raising cytosolic levels of the metabolite *S*-adenosyl methionine (SAM). Consistent with 2-DOG treatment, explants incubated with SAM between E6.5 and E13.5 lacked correct tonotopic patterning indicated by the flattening of HC morphologies along the tonotopic axis (*Figure 6—figure supplement 3*).

## Flux through the PPP is important for tonotopic patterning along the BP

Studies in other systems have linked both the PPP and TCA cycle with cell fate decisions during development and differentiation (*Chi et al., 2020*; *Arnold et al., 2022*). We therefore sought to further dissect the metabolic signalling network during specification of tonotopy in the developing BP. To investigate a role for PPP-linked glucose metabolism, BP explants were established as described, and treated with 50 mM 6-aminonicotinamide (6-AN) between E6.5 and E13.5 equivalent. Treatment with 6-AN inhibits the rate-limiting PPP enzyme glucose 6-phosphate dehydrogenase (*Figure 1*). By comparison with control cultures, inhibition of PPP metabolism caused a significant decrease in HC size within 2500 µm² areas measured in the proximal BP (*Figure 7*, *Figure 7—figure supplement 1*).

To determine whether this effect was specific to glucose flux through the PPP we also blocked phosphofructokinase (PFK), a rate-limiting enzyme further down in the glycolytic pathway, using 1 mM YZ9 (*Figure 7—figure supplement 2*). Blocking PFK activity inhibits the glycolytic cascade involved in pyruvate production but does not change the activity of G6PD in the PPP (*Chi et al., 2020*). YZ9 treatment resulted in a reduction in HC size, especially in the proximal region leading to a reduction in the HC gradient when analysed using pairwise comparisons (Sidak's multiple comparisons, p = 0.17). However, YZ9 was unique among our metabolic inhibitor treatments in that it did not produce a significant interaction term in our two-way analysis of variance (ANOVA) (*Figure 7—figure supplement 2C* and *Source data 1* and *Source data 2*). 2-DOG, SAM, and 6-AN treatments conversely all produced significant interaction terms after treatment, indicating a reduction in the normal proximal–distal gradient in cell size similar to that observed after Chdl1 and Bmp7 treatments (*Source data 1* and *Source data 2*). Therefore, whilst blocking glycolysis and the PPP elicited significant changes in the cell size, disrupting the downstream branches of glycolysis while leaving PPP flux intact had no effect (*Figure 7—figure supplements 1 and 2* and *Source data 1* and *Source data 2*). Consistent with a higher NADPH/NADH ratio these findings suggest that tonotopic patterning is regulated by glucose flux in the upper branch of glycolysis, rather than by enzymes in the lower branch of the pathway.

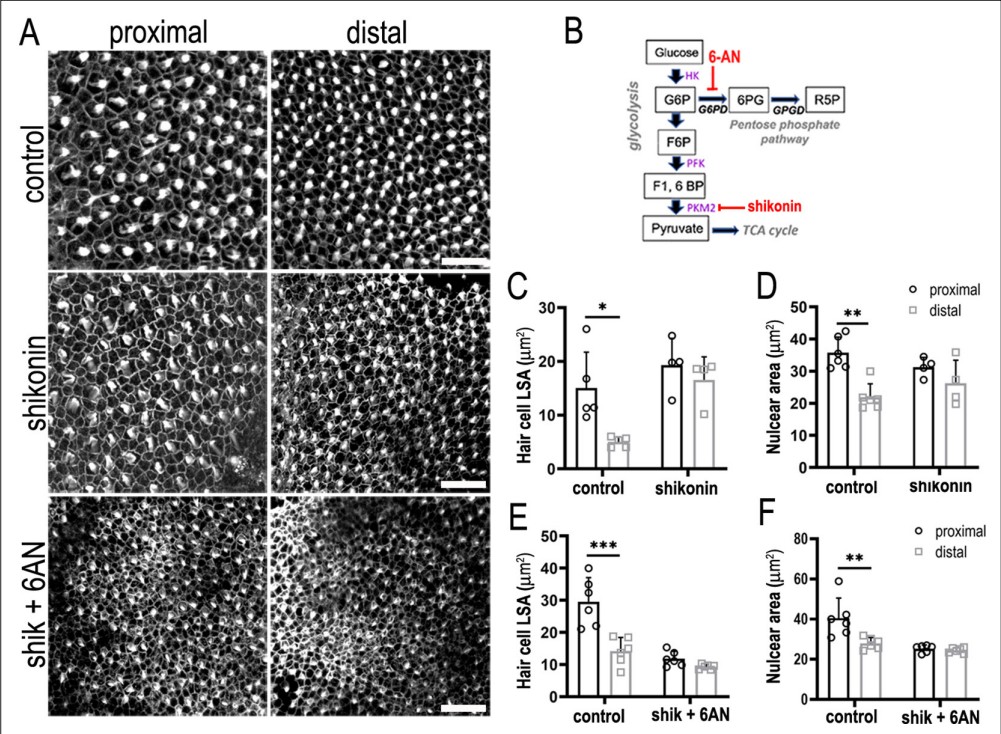

**Figure 8.** Glucose catabolism in pentose phosphate pathway (PPP) and Pkm2 pathways is necessary for specifying correct hair cell (HC) morphology and patterning along the tonotopic axis of the developing basilar papilla (BP). (**A**) Maximal z-projections showing Phalloidin staining at the epithelial surface in proximal and distal BP regions of control and shikonin and shikonin + 6-aminonicotinamide (6-AN)-treated explants. (**B**) Schematic illustrating the metabolic enzymes targeted by shikonin and 6-AN. (**C, D**) Quantification of HC lumenal surface area and nuclear are in control and shikonin BP explants established at E6.5 and maintained for 7 days in vitro. (**E, F**) Quantification of HC lumenal surface area and nuclear are in control and shikonin + 6 AN explants treated from E6.5 for 7 days in vitro. HC circumference, nuclear area, and HC density were quantified in 2500 μm² regions of interest (ROIs) in proximal and distal regions. Data are mean ± standard error of the mean (SEM). *p = <0.05, **p = <0.01, ***p = <0.001 two-way analysis of variance (ANOVA). Controls: n = 5 for LSA, n = 6 for nuclear area, shikonin: n = 4, shikonin + 6-AN: n = 6. Scale bars are 20 μm.

## Combined activity in PPP and Pkm2 pathways is required for specification of HC positional identity and patterning in the developing BP sensory epithelium

Having identified graded differences in Pkm2 expression (*Figure 3—figure supplement 3*, *Figure 4*), we investigated whether the enzyme is required for specification of HC morphologies during development. Although both Pfk and Pkm2 regulate glycolysis, the two enzymes regulate different stages of the pathway with respect to glucose flux and the re-entry of PPP products. Inhibition of Pkm2 would cause build-up of both G3p and F6p, whereas inhibition of PFK would only lead to accumulation of F6P. We therefore hypothesised that Pkm2 inhibition would have a greater impact PPP activity and tonotopic patterning.

Pkm2 activity was blocked during development from E6.5 to E13.5 equivalent using the pharmacological inhibitor shikonin (*Figure 8*). Treatment of explants with shikonin during HC formation abolished the normal tonotopic gradient in HC lumenal surface area (*Figure 8A, C*), where HC cuticular plate circumference was significantly increased in the distal but not proximal BP region compared to controls. Absence of Pkm2 activity did not alter HC nuclear area (*Figure 8D*), suggesting that the enzyme could play a more important role regulating HC morphology and patterning at the epithelial surface.

2-DOG blocks the activity of all pathways in cytosolic glycolysis, a network that encompasses both Pkm2 and PPP-linked glucose catabolism. Having probed Pkm2 and PPP metabolism independently

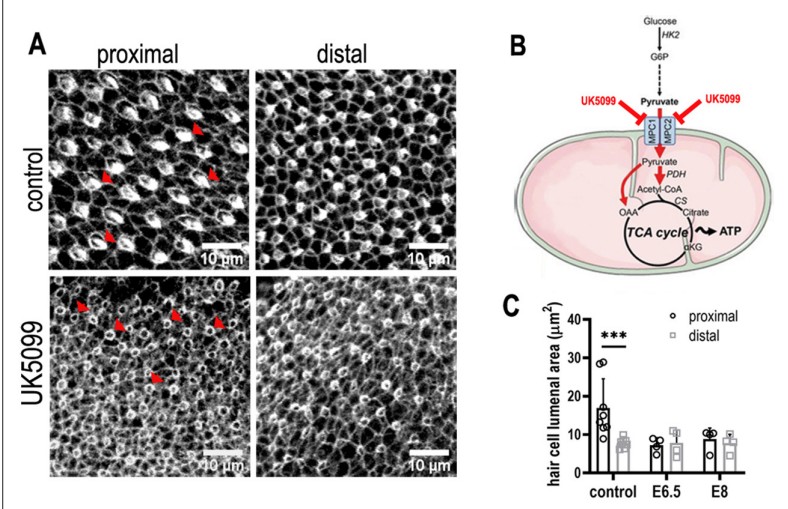

**Figure 9.** Mitochondrial OXPHOS is necessary for the normal developmental progression of hair cells (HCs) but not positional identity. (**A**) HC morphology at the surface of the basilar papilla (BP) epithelium in explants stained with Phalloidin. Cultures were established at E6.5 and maintained for 7 days in vitro in control medium or that supplemented with the mitochondrial inhibitor UK5099. (**B**) Blocking pyruvate uptake into mitochondria with UK5099 disrupts normal tricarboxylic acid (TCA) cycle activity and thus mitochondrial OXPHOS by blocking pyruvate uptake via the mitochondrial pyruvate carrier (MPC1/MPC2). (**C**) Blocking mitochondrial OXPHOS from E6.5 to E13.5 equivalent caused developmental abnormalities in HCs along the BP including reduced HC size and immature stereocilial bundles (red arrowheads in A) in both proximal and distal regions compared to controls. To test whether mitochondrial OXPHOS is required for overall developmental progression, cultures were also established at E8 and treated with 1 µM UK5099 for 7 days in vitro to the developmental equivalent of E15. HCs showed no apparent recovery of normal morphology following UK5099 treatment from E8 compared to E6.5. Data are mean ± standard error of the mean (SEM). ***$p < 0.001$, two-way analysis of variance (ANOVA). Controls: $n = 8$, UK5099 E6.5: $n = 4$, E8: $n = 4$.

The online version of this article includes the following figure supplement(s) for figure 9:

**Figure supplement 1.** Blocking mitochondrial OXPHOS delays developmental acquisition of normal hair cell (HC) morphologies.

we tested whether blocking both pathways during development was additive and could mimic the inhibitory effects of 2-DOG on HC positional identity (*Figure 6*). We find that blocking both pathways during development abolished the normal gradient in HC morphology along the tonotopic axis, thus mimicking the effects seen when inhibiting glycolysis using 2-DOG (*Figure 8A, E, F*). These data therefore suggest that metabolic activity in both PPP and Pkm2 pathways is important for establishing HC positional identity along the developing BP.

## Pyruvate-mediated OXPHOS in mitochondria maintains progression of HC development but does not regulate tonotopic patterning

In the developing BP, live imaging of mitochondrial activity using TMRM revealed no apparent difference in OXPHOS along the tonotopic axis. To determine whether mitochondrial metabolism influences tonotopic patterning during development, we blocked uptake of glycolytically derived pyruvate into mitochondria by inhibiting the mitochondrial pyruvate carrier with UK5099 (*Zhong et al., 2015*; *Figure 9B*). HCs in explants treated with UK5099 between E6.5 and E13.5 developed with abnormal morphologies at all positions along the BP and displayed either immature stereociliary bundles or lacked them completely (*Figure 9A, C*, red arrowheads). To determine whether this effect was due to a global developmental arrest, explants were also established at E8, by which time tonotopy is specified (*Mann et al., 2014*) but bundles are not yet developed, and maintained for 7 DIV to the equivalent of E15. Compared to controls, HCs of explants treated with UK5099 were smaller and displayed abnormal bundle morphologies at all positions along the BP (*Figure 9C*, *Figure 9— figure supplement 1*). The role that mitochondria play in shaping HC morphologies and functional properties at different frequency positions is at present unclear and will require further investigation.

However, our findings suggest that pyruvate-mediated OXPHOS plays a more significant role in maintaining the overall progression of development rather than regulating the patterning along the tonotopic axis.

## Glucose metabolism regulates expression of *Bmp7* and *Chdl1* along the tonotopic axis

In many developing systems, gradients of one or more morphogen act to regulate cell fate, growth and patterning along a given axis (*Towers et al., 2012*; *Averbukh et al., 2014*). In the chick cochlea, reciprocal gradients of *Bmp7* and its antagonist *Chdl1* play key roles in determining HC positional identity. As disruption of both the normal gradient in *Bmp7* (*Mann et al., 2014*) and glucose metabolism induce similar defects in morphological patterning (*Figure 6*, *Figure 10A–D*), we investigated the possibility of a causal interaction between the metabolic and the morphogen signalling networks in the developing BP.

To investigate the regulatory effects of cytosolic glucose metabolism on the expression gradients of *Bmp7* and *Chdl1*, explants were established at E6.5 and maintained for 72 hr in vitro (equivalent of E9.5) in control medium or that containing 2-DOG + NaP. Whole-mount in situ performed on explant cultures showed that disrupted glucose metabolism altered the normal expression of *Bmp7* and *Chd1*along the BP (*Figure 10E*). Following treatment with 2-DOG, *Bmp7* expression appeared to increase along the entire BP while *Chdl1* showed a reciprocal decrease along the length of the organ. It is challenging to predict how this global change in expression levels would impact the activity of each morphogen along the tonotopic axis, but it does support our hypothesis that there is a causal interaction between glycolytic and Bmp7-Chdl1 networks. The precise nature of this interaction requires further investigation. We speculate that the increased *Bmp7* and reduced *Chdl1* expression in the proximal region (*Figure 10E*, *Figure 10—figure supplement 1*), in response to perturbed glucose flux (by treating with 6-AN), would induce expansion of distal-like HC morphologies into the proximal region. Blocking mitochondrial-linked glucose catabolism with UK5099 did not alter the expression of *Bmp7* (*Figure 10—figure supplement 2*).

## Treatment with Chdl1 restores normal tonotopic patterning when glycolysis is blocked during development

Modulating the reciprocal gradients of *Bmp7* and *Chdl1* along the proximal-to-distal axis alters tonotopic patterning in nascent HCs (*Mann et al., 2014*). We further show that the normal gradients of *Bmp7* and *Chdl1* are disrupted along the BP when glycolysis or PPP activity are blocked during development (*Figure 10E*, *Figure 10—figure supplement 1*). As treatment with 2-DOG causes a global increase in *Bmp7* and decrease in *Chdl1* expression (*Figure 10E*), we therefore hypothesised that treatment of explants with 2-DOG in the presence of Chdl1 protein might restore tonotopic patterning when glycolysis is blocked. Analysis of explants treated between E6.5 and E13.5 equivalent with 2-DOG + 0.4 µg/ml Chdl1 showed a partial rescue of HC morphologies (lumenal surface area and nuclear area) along the proximal-to-distal axis (*Figure 11*, *Figure 11—figure supplement 1*, and *Source data 1* and *Source data 2*). Whilst the precise length and number of stereocilia could not be accurately quantified using Phalloidin staining in these explants, the overall bundle morphology also appeared consistent with that reported previously for proximal and distal BP regions (*Thiede et al., 2014*; *Tilney and Saunders, 1983*; *Son et al., 2015*).

Taken together, our findings suggest that a distinct metabolic state coupled with a specific morphogen level can regulate HC morphology at different positions along the tonotopic axis during development. These data also provide further evidence indicating a causal interaction between metabolic and morphogen signalling networks during development. Ascertaining a role for cytosolic glucose metabolism in specifying proximal verses distal HC fate, specifically related to frequency tuning, would require a detailed analysis of HC physiological properties. Future work should therefore determine whether altering metabolism affects the developmental acquisition of not only HC morphology, but also the intrinsic electrophysiological properties and firing characteristics documented for HCs at different tonotopic positions (*Fuchs and Evans, 1990*; *Fuchs et al., 1988*; *Fettiplace and Fuchs, 1999*).

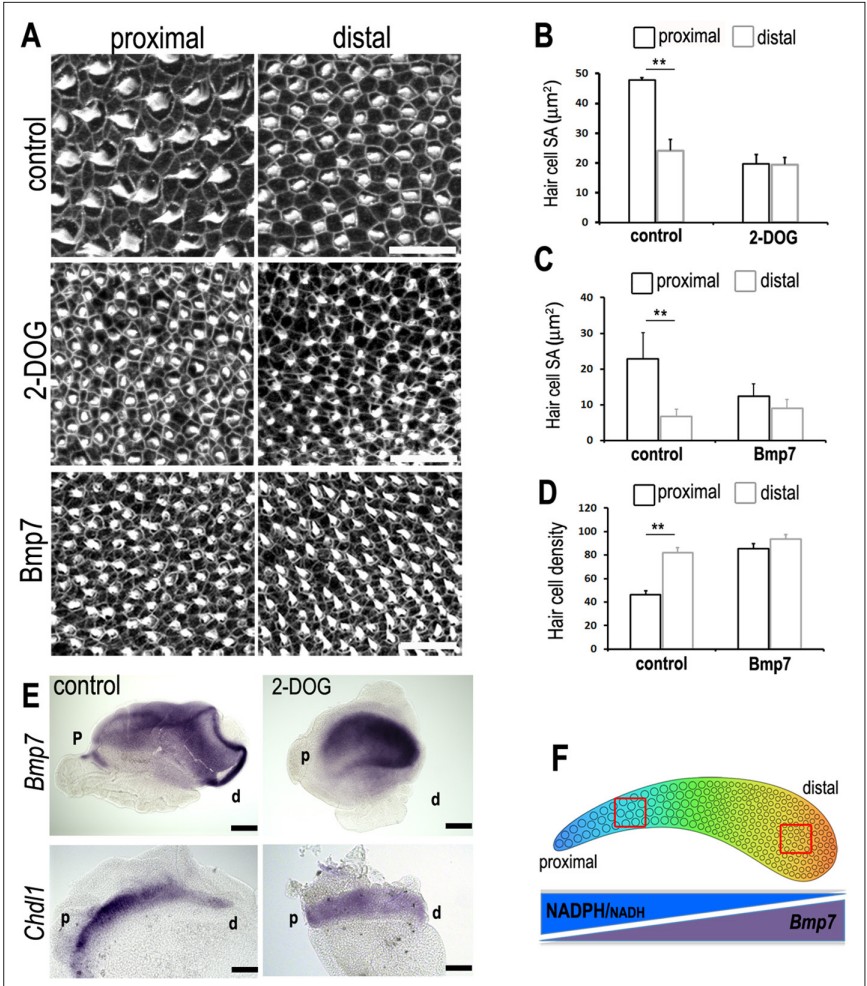

**Figure 10.** A tonotopic gradient in NAD(P)H producing glucose metabolism specifies hair cell (HC) positional identity along the basilar papilla (BP) by regulating gradients of *Bmp7* and *Chdl1*. (**A**) Phalloidin staining at the surface of BP explants in the proximal and distal regions. Explants were established at E6.5 and incubated for 7 days in vitro in control medium or medium containing 2-deoxy-D-glucose (2-DOG) + sodium pyruvate (NaP) or Bmp7. (**B, C**) Treatment with 2-DOG- or Bmp7-induced HC morphologies consistent with a more distal phenotype in the proximal BP. HC lumenal surface area was determined using Phalloidin staining at the cuticular plate in 2500 μm² areas. (**D**) Treatment with Bmp7 between E6.5 and E13.5 equivalent results in increased HC density in the proximal BP region. HC density was counted in proximal and distal BP regions using defines regions of interest (ROIs) of 10,000 μm². (**E**) Treatment of explant cultures with 2-DOG + NaP from E6.5 for 72 hr in vitro disrupts the normal tonotopic expression of *Bmp7* and its antagonist *Chdl1*. Images show in situ hybridisation for *Bmp7* and *Chdl1* in BP whole-mounts treated with 2-DOG + NaP from E6.5 for 72 hr in vitro. Images are representative of 6 biological replicates. (2-DOG) Controls: *n* = 6, 2-DOG: *n* = 6. Data mean ± standard error of the mean (SEM). **p < 0.01 two-way analysis of variance (ANOVA). (Bmp7) Controls: *n* = 11, Bmp7: *n* = 10. Data mean ± SEM. **p < 0.01 two-way ANOVA. Scale bars (**A**) control scale bar is 20 μm, DOG and Bmp7 are 50 μm. Scale bars for in situ data (**E**) are 10 μm. (**F**) Schematic of the chick BP, showing the graded differences in HC size and density along the tonotopic axis. The opposing gradients in *Bmp7* activity and in cellular NAD(P)H/NADH (glycolysis) are indicated. Red boxes indicate regions of measurement for HC lumenal surface areas and cell density.

The online version of this article includes the following figure supplement(s) for figure 10:

**Figure supplement 1.** Blocking glucose flux into the pentose phosphate pathway (PPP) with 6-aminonicotinamide (6-AN) alters the normal expression of *Bmp7* and *Chdl1* in the developing basilar papilla (BP).

**Figure supplement 2.** Blocking mitochondrial metabolism does not alter the gradient of *Bmp7* expression along the developing basilar papilla (BP).

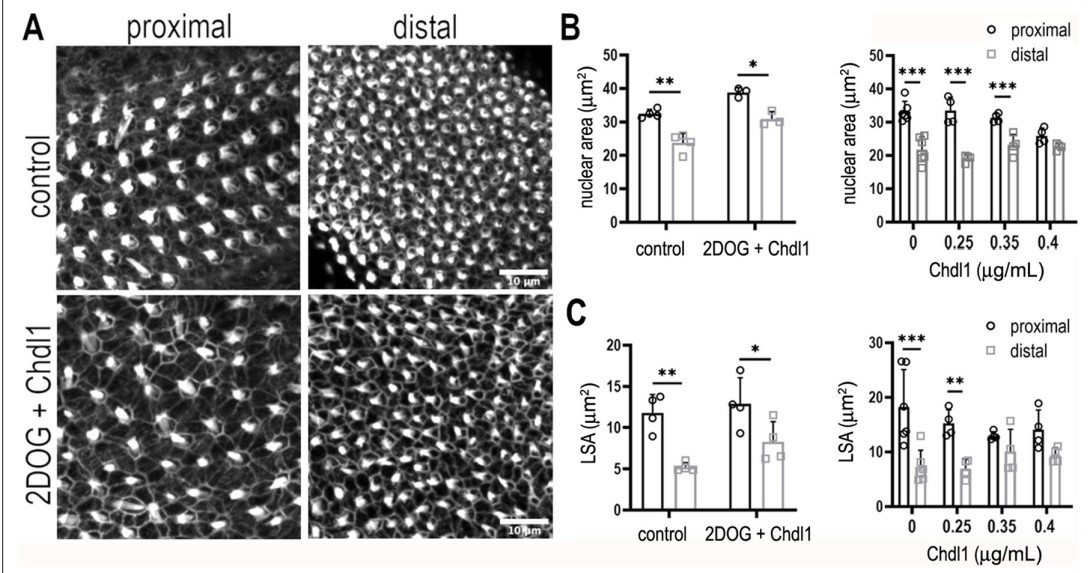

**Figure 11.** Chdl1 restores tonotopic patterning along the basilar papilla (BP) when cytosolic glycolysis is blocked with 2-deoxy-D-glucose (2-DOG).
(**A**) Maximum z-projections of the epithelial surface in the proximal and distal regions of Phalloidin-stained BPs. Explants were established at E6.5 and maintained for 7 days in vitro in either control medium or that containing 2-DOG and Chdl1. Phalloidin staining indicates differences in hair cell (HC) lumenal surface area and gross bundle morphology between proximal and distal regions. The tonotopic gradient in HC lumenal surface was restored when explants were treated with 2-DOG in the presence of Chdl1. (**B**) The gradient in HC nuclear area was maintained along the proximal-to-distal axis when explants were treated with 2-DOG in the presence of Chdl1. Effects of Chdl1 were only apparent at concentrations of 0.35 µg/ml or above. (**C**) HC lumenal surface area measured from 2500 µm² regions of interest (ROIs) in proximal (black) and distal (grey) BP regions. Effects of different Chdl1 concentrations on the HC luminal surface are indicated. Data are mean ± standard error of the mean (SEM). ** $p < 0.01$, *** $p < 0.001$ two-way analysis of variance (ANOVA). Dose–response data – Controls: $n = 6$, Chdl1 0.25 $n = 4$, 0.35 $n = 4$, 0.4 $n = 4$. Chdl1 + 2-DOG: Controls $n = 6$, Chdl1 + 2-DOG $n = 4$ biological replicates for HC lumenal surface areas and $n = 3$ for nuclear area. Scale bars are 10 µm.

The online version of this article includes the following figure supplement(s) for figure 11:

**Figure supplement 1.** Treatment of explants with 2-deoxy-D-glucose (2-DOG) + Chdl1 partially rescues the pattern of hair cell (HC) density along the developing basilar papilla (BP).

## Discussion

Generating new HCs that recapitulate the features of those in a healthy cochlea requires a detailed knowledge of the cell biology driving their formation. As high-frequency HCs are more vulnerable to insult, there is also a need to understand differences in the specific factors and signalling pathways that drive the identity of distinct HC subtypes. Over recent years, metabolism has emerged as a key driver of cell fate and function across various biological systems and cellular contexts (*Ghosh-Choudhary et al., 2020*; *Tarazona and Pourquié, 2020*). Taking both qualitative and quantitative approaches, we identify regional differences in metabolism along the frequency axis of the developing chick cochlea and explore a role for causal signalling between graded morphogens and glucose metabolism in establishing HC positional identity. We identify a tonotopic gradient in cellular NADPH, originating from differences in glucose flux between high- and low-frequency HCs. Tonotopic differences in the catabolic fate of glucose in glycolysis or the PPP modulates Bmp7 and Chdl-1 signalling along the developing BP. This study provides the first evidence supporting a role for crosstalk between metabolism and morphogen gradients in the developing auditory system, building on our current understanding of cell fate specification.

### NAD(P)H FLIM reveals a gradient in metabolism along the tonotopic axis of the developing chick cochlea

Using NAD(P)H FLIM, we uncovered a proximal-to-distal gradient in cellular NADPH resulting from tonotopic differences in the fate of cytosolic glucose. The biochemical basis for this gradient was further investigated by exploring differences in mitochondrial activity, $pH_i$, and the expression of metabolic enzymes along the tonotopic axis. Collectively, these analyses indicate that differences in

the fate of cytosolic glucose, (*Grüning et al., 2011*; *Yang and Lu, 2015*; *Alfarouk et al., 2020*) rather than between glycolytic and oxidative metabolic states underpin the increased $\tau_{bound}$ lifetime and higher NADPH/NADH ratio reported here for high-frequency HCs. The higher expression of PKM2 and the more alkaline pH identified in proximal HCs are also consistent with a higher cellular NADPH/NADH ratio (*Nandi et al., 2020*; *Zhang et al., 2019*).

## PPP metabolism, HC size, and positional identity

Cell geometry and size contribute to overall tissue architecture during development and are important for long-term function of the cochlea in vertebrates. Cell size, cell membrane composition, and metabolic rate are tightly correlated (*Hulbert and Else, 1999*). PPP-derived NADPH is utilised extensively in proliferating cells during development, where it regulates cell cycle progression, differentiation, and growth (*Vander Heiden et al., 2009*; *Vizán et al., 2009*). Increased glucose metabolism has been reported in regenerating (*Love et al., 2014*; *Patel et al., 2022*) and developing systems (*Chi et al., 2020*; *Oginuma et al., 2017*; *Bulusu et al., 2017*; *Miyazawa et al., 2022*) where it plays an important role in regulating cell fate, behaviour, and shape. However, the chosen path of glucose catabolism is context dependent and differs across processes and between tissues. PPP metabolism regulates cell division and proliferation through its ability to generate lipid and nucleotide precursors (*Stincone et al., 2015*). Increased glucose flux into the PPP but not the main branch of glycolysis was also recently shown to regulate the balance between proliferation and cell death in the regenerating limb (*Patel et al., 2022*). In the BP, HCs exit the cell cycle in three progressive waves following a centre-to-periphery progression beginning at E5. During this process, both HCs and SCs are added in apposition, to the edges of the band of post-mitotic cells that preceded them (*Katayama and Corwin, 1989*). HC differentiation then begins in the distal portion of the BP at around E6 and extends proximally along the cochlea expanding across the width of the epithelium (*Goodyear and Richardson, 1997*; *Cotanche and Sulik, 1984*; *Bartolami et al., 1991*). These graded differences in HC size along the organ are an essential requirement for correct auditory coding (*Fettiplace and Fuchs, 1999*). As larger cell size is correlated with increased G6PDH activity and thus more glucose flux into the PPP (*Vizán et al., 2009*), the higher activity in the proximal BP region may underlie the graded differences in HC size that arise during development. It could be argued that the smaller cell size induced following metabolic perturbation in the proximal BP is a result of impaired differentiation. However, given that distal HCs maintain their small size in the mature organ, this morphological change observed in the proximal region following treatment with 2-DOG and 6-AN is consistent with a distal-like phenotype. Without a detailed characterisation of distal and proximal cell growth from E6 through to adult stages, this puzzle is challenging to resolve.

PPP metabolism is also closely linked with de novo synthesis of lipids and cholesterol, which form an integral part of cell membranes (*Sykes et al., 1986*). Functional interactions between ion channel complexes in the membrane and the local lipid environment have been described previously in mammalian (*Hirono et al., 2004*; *Gianoli et al., 2017*) and avian HCs (*Purcell et al., 2011*). Frequency tuning in the BP relies on the intrinsic electrical properties of the HCs themselves, where graded differences in the number and kinetics of voltage-gated calcium channels and calcium-sensitive (BK type) potassium channels underlie the ability of HCs to resonate electrically in response to sound (*Fettiplace and Fuchs, 1999*). By regulating cholesterol in the HC membrane, the graded PPP activity observed here may also regulate aspects of HC electrical tuning.

## A causal link between metabolism and morphogen signalling during development sets up HC positional identity

Morphogen signalling gradients have well defined roles in directing cell identity along developing axes, where cells determine their fate as a function of morphogen concentrations at different positions along them (*Towers et al., 2012*; *Dessaud et al., 2008*; *Sagner and Briscoe, 2019*). We showed previously that reciprocal gradients of Bmp7 and Chdl1 establish HC positional identity along the developing BP (*Mann et al., 2014*). The gradient of Bmp7 is established by Sonic hedgehog (Shh) signalling emanating from ventral midline structures, including the notochord and floor plate (*Son et al., 2015*). Here, we identify a gradient in glucose metabolism that regulates the morphology of developing HCs along the tonotopic axis through a causal interaction with the Bmp7–Chdl1 network. Disrupting this gradient using 2-DOG, SAM, or 6-AN mimics the effects on HC morphology reported

previously for altered Shh (*Son et al., 2015*) and Bmp7 signalling (*Mann et al., 2014*), which induced distal-like HC phenotypes in the proximal BP. Furthermore, the effects of impaired glycolysis could be partially rescued in explants treated with 2-DOG in the presence of Chdl1. Our findings thus indicate a complex and causal interplay between Bmp7 and Chdl1 morphogen gradients and glucose metabolism in the specification of HC tonotopic identity. Metabolic gradients are also known to regulate elongation of the body axis and somite patterning (*Oginuma et al., 2017*; *Miyazawa et al., 2022*). By establishing a gradient in intracellular pH, glycolysis drives graded Wnt signalling and specifies mesodermal versus neuronal cell fate along the developing body axis (*Oginuma et al., 2017*; *Oginuma et al., 2020*; *Bulusu et al., 2017*). The specific role of the $pH_i$ gradient along the developing BP is unclear, however given the importance of the Wnt signalling pathway in both cochlear development and regeneration and repair, it will be important to investigate crosstalk between the two in the context of HC formation and tonotopic identity.

In conclusion, our findings indicate a causal link between PPP activity and graded morphogen signalling in specifying HC morphology along the tonotopic axis during development. However, a detailed physiological analysis is required to accurately confirm whether these morphological changes reflect a switch between proximal and distal HC fate. Future work should determine how altering the fate of glucose affects the morphological and functional development of the stereociliary bundle, the intrinsic electrophysiological properties and the firing characteristics of HCs at different tonotopic positions. Untangling further the interactions between the components of the Shh, Bmp7, and metabolic signalling networks will advance our understanding of how HCs acquire the unique morphologies necessary for auditory coding. From what we understand about frequency selectivity in vertebrates (*Gleich et al., 2005*), recapitulation of tonotopy will require that any gradient, and its associated signalling networks, scale correctly in different inner ear sensory patches and across species with varying head size and cochlear lengths. Understanding how the mechanical constraints associated with growth and patterning in different sense organs modulate these networks will advance our understanding of how to drive formation of specific HC phenotypes in inner ear organoid models.

## Materials and methods
### Embryo care and procedures
Fertilised White Leghorn chicken (*Gallus gallus domesticus*) eggs (Henry Stewart & Co Ltd, UK) were incubated at 37.5°C in an automatic rocking incubator (Brinsea) until use at specific developmental stages between embryonic day 6 (E6) and E16. Embryos were removed from their eggs, staged according to Hamburger and Hamilton (1951) and subsequently decapitated. All embryos were terminated prior to hatching at E21. All procedures were performed in accordance with United Kingdom legislation outlined in the Animals (Scientific Procedures) Act 1986.

### Preparation of BP explants for live imaging studies
BPs were collected from chick embryos between E7 and E16, and explants were established at E13 to E16 in accord with United Kingdom legislation outlined in the Animals (Scientific Procedures) Act 1986. Explants were placed onto Millicell cell culture inserts (Millipore ) and maintained overnight at 37°C in medium 199 Earl's salts (M199) (Gibco, Invitrogen) containing 2% foetal bovine serum and 5 mM 4-(2-hydroxyethyl)-1-piperazineethanesulfonic acid (HEPES) buffer (Life Technologies). For live imaging experiments, cultures were transferred to glass-bottom 50 mm MatTek dishes and held in place using custom-made tissue harps (Scientifica). Cultures were maintained in L-15 medium at room temperature throughout the experiment.

### BP culture
BPs were isolated from embryos incubated for between 6 (E6.0) and 8 (E8.0) days and maintained in chilled Leibovitz's L-15 media (Gibco, Invitrogen). Papillae were dissected as described previously (*Jacques et al., 2014*) and cultured nerve-side-down on Millicell cell culture inserts (Millipore ). Cell culture inserts were placed into 35-mm culture dishes containing 1.5 ml of 199 Earl's salts (M199) medium (Gibco, Invitrogen) supplemented with 5 mM HEPES buffer and 2% foetal bovine serum. Papillae were maintained in M199 medium plus vehicle (control media) for up to 7 DIV until the equivalent of E13.5. For all treatments, a minimum of four samples were analysed. The following

factors were applied to experimental BPs in culture at the specified concentrations: 2-deoxyglucose (2-DOG) 2 mM (Sigma), NaP 5 mM (Sigma), 6-AN 2 μM (Sigma), S-(5'-adenosyl)-L-methionine chloride dihydrochloride (SAM) 50 μM (Sigma), YZ9 1 μM (Sigma), Shikonin 1 μM (Sigma), Bmp7 recombinant protein 0.4 μg/ml (R&D Systems 5666-BP-010/CF), and Chordin like-1 recombinant protein 0.4 μg/ml (R&D Systems 1808-NR-050/CF). For 2-DOG wash-out experiments, cultures were treated for 24 or 48 hr followed by wash out with control medium for the remainder of the experiment up to 7 days. For paired controls, medium was also changed at 24 and 48 hr in culture. At the conclusion of each experiment (7 DIV), cultures were fixed in 4% paraformaldehyde (PFA) for 20 min at room temperature, washed thoroughly three times with 0.1 M phosphate buffered saline (Invitrogen) and processed for immunohistochemistry.

## Fluorescence lifetime imaging

NAD(P)H FLIM was performed on an upright LSM 510 microscope (Carl Zeiss) with a 1.0 NA ×40 water-dipping objective using a 650 nm short-pass dichroic and 460 ± 25 nm emission filter. Two-photon excitation was provided by a Chameleon (Coherent) Ti:sapphire laser tuned to 720 nm, with on-sample powers kept below 10 mW. Photon emission events were registered by an external detector (HPM-100, Becker & Hickl) attached to a commercial time-correlated single photon counting electronics module (SPC-830, Becker & Hickl) contained on a PCI board in a desktop computer. Scanning was performed continuously for 2 min with a pixel dwell time of 1.6 μs. Cell type (HC vs. SC) and z-position within the epithelium was determined prior to FLIM analysis using the mitochondrially targeted fluorescent dye TMRM. The dye was added to the recording medium, at a final concentration of 350 nM, 45 min before imaging. TMRM fluorescence was collected using a 610 ± 30 nm emission filter. Excitation was provided at the same wavelength as NAD(P)H to avoid possible chromatic aberration. The 585 ± 15 nm emission spectrum of TMRM ensured its fluorescence did not interfere with acquisition of the NAD(P)H FLIM images.

## FLIM data analysis

Following 5 × 5 binning of photon counts at each pixel, fluorescence decay curves of the form (*Blacker et al., 2014*),

$$I(t) = Z + I_0([1 - \alpha_{bound}]e^{-t/\tau_{free}} + \alpha_{bound}e^{-t/\tau_{bound}})$$

were fit to the FLIM images using iterative reconvolution in SPCImage (Becker & Hickl), where $Z$ allows for time-uncorrelated background noise. Matrices of the fit parameters $\tau_{free}$, $\alpha_{bound}$, and $\tau_{bound}$ and the total photons counted were at each pixel, were exported and analysed for HCs and SCs, and proximal and distal BP regions, using SPCImage and ImageJ software packages.

## 2-NBDG, TMRM live imaging

BP were isolated from E7, E9, E14, and E16 chick embryos in chilled L-15 mediumand subsequently incubated in 1 mM solution of 2-NBDG (N13195, Thermo Fisher Scientific) in L-15 medium at room temperature for 1 hr. The medium was then replaced with a fresh solution of 1 mM 2-NBDG and 350 nm TMRM (T668, Thermo Fisher Scientific) in L-15 and incubated for a further hour at room temperature. Thereafter, the BPs were washed several times with fresh medium containing 350 nM TMRM and mounted in a 3.5-mm glass bottom MatTek dish. 3D image stacks with an optical thickness of 1 μm were captured using a Leica SP5 confocal microscope with an HCX PL APO ×63/1.3 GLYC CORR CS (21°C) objective.

## Measurement of pH$_i$ using pHrodo Red

BP was dissected in cold L-15 media and incubated for 1 hr at room temperature with 5 μM pHrodo Red Intracellular pH Indicator (Invitrogen P35372) and 1 nM SiR-actin (SpiroChrome SC001) in L-15 medium. Samples were subsequently mounted in Mattek (50 mm) dishes and held in place using custom-made imaging grids. Explants were imaged using an inverted ZEISS LSM980 confocal microscope using a ×63 objective and digital zoom of ×1.8. Z-intervals were kept consistent at 0.4 μm across all developmental stages.

## Immunohistochemistry

Inner ear tissue was collected at various developmental stages, fixed for 20 min to 1 hr in in 0.1 M phosphate-buffered saline (PBS) containing 4% PFA, and processed for whole-mounts immunohistochemistry. The BP was then fine dissected and permeabilised in PBS containing 0.3% Triton for 30 min before immunostaining using standard procedures (*Mann et al., 2014*). Samples were stained with primary antibodies for PKM2 1:100 (Cell Signalling 4053T). Antibody staining was visualised using secondary antibodies conjugated to either Alexa 488 or Alexa 546 (Invitrogen). Cultures were incubated with all secondary antibodies for 1 hr at room temperature 1:1000, washed thoroughly in 0.1 M PBS. Samples were then incubated for an additional hour with either Alexa Fluor-conjugated Phalloidin-546 1:250 (Invitrogen) to label filamentous actin and DAPI 1:1000 to label cell nuclei. Tissue samples were mounted using Prolong Gold antifade reagent (Invitrogen). 3D image stacks of mounted samples were captured using a Leica SP5 confocal microscope with an HCX PL APO ×63/1.3 GLYC CORR CS (21°C) objective.

## EdU staining

Control or 2-DOG-treated cultures were incubated for 48 hr in 10 μM EdU from E8 to E10. Cultures were subsequently fixed for 15 min in 4% PFA at room temperature and then washed in 0.1 M PBS. Explants were then processed for EdU staining following the Click-iT EdU 488 protocol (Thermo Fisher Scientific).

## Image analysis

Analysis of z-stacks from immunohistochemistry stains as well as 2-NBDG, TMRM, and pHrodo Red live imaging experiments was carried out using the Fiji distribution of ImageJ. For each sample, a z-plane 2 μm beneath the surface of the epithelium was selected using Phalloidin or SiR-actin labelling for further analysis. For each of these selected z-planes, a 100 μm × 100 μm ROI was chosen containing intact tissue in which all HCs were optimally orientated for analysis. Mean fluorescence intensity of the tissue was measured for HCs and SCs from within defined 100 μm × 100 μm ROIs at E7, E9, and E10 timepoints. At E14 and E16, HCs and SCs were manually segmented. At younger stages, when HCs and SCs were not easily identified, fluorescence intensity was measured from within the whole epithelium. HC labels were dilated by 3 μm, which provided selections which included both HCs and their surrounding SCs. By subtracting the HC segmentation from the dilated label, we were thus able to measure the fluorescence intensity of whole HCs separately from their surrounding support cells in the 2-NBDG data. A similar approach was adopted when measuring TMRM fluorescence intensity at E14 and E16. However, we noticed that signal was concentrated around the HC periphery. To ensure that the fluorescence intensity best reflected only the mitochondria and was not reduced by the low fluorescence from the centre of each HC, we measured mean fluorescence intensity only up to 2 μm from the cell membrane. Likewise, for TMRM data at E7 and E9, mitochondria were segmented using Fiji's auto-local thresholding (Niblack) prior to intensity measurements, to avoid a biased estimate of fluorescence intensity due to empty space surrounding each mitochondrion.

## Analysis of HC morphology

Data were analysed offline using ImageJ software. HC lumenal surface area and cell size were used as indices for HC morphology along the tonotopic axis. To determine the HC density, the lumenal surfaces of HCs and cell size, cultures were labelled with Phalloidin and DAPI. Then, the number of HCs in 50 μm × 50 μm ROI (2500 μm² total area) located in the proximal and distal BP regions were determined. Proximal and distal regions were determined based on a calculation of the entire length of the BP or explant. In addition, counting ROIs were placed in the mid-region of the BP along the neural to abneural axis to avoid any confounding measurements due to radial differences between tall and short HCs. For each sample, HCs were counted in four separate ROIs for each position along the BP. Lumenal surface areas were determined by measuring the circumference of individual HCs at the level of the cuticular plate. Nuclear size was determined using the DAPI signal.

## Statistical testing and analyses

All data were assessed for normality prior to application of statistical tests, with a threshold of $p < 0.05$ used for determining significance. When comparing between proximal and distal regions within

the same tissue explant, paired *t*-tests with unequal variance were used. This statistical approach was chosen given that measurements were made from different regions within the same sample and were therefore not independent from each other. Comparisons made between different developmental stages were assumed independent from one another and thus here, independent *t*-tests and two-way ANOVAs were used.

## In situ hybridisation

Inner ear tissue was dissected and fixed in 4% PFA overnight at 4°C. Tissue was subsequently washed three times for 30 min in 0.1 M PBS, dehydrated in ascending methanol series (25–100%) and stored at −20°C until use. Immediately before the in situ protocol, tissue was rehydrated in a descending methanol series (100–25%). Antisense digoxigenin-labelled RNA probes for *Bmp7* were kindly provided by Doris Wu (NIDCD, NIH). *Chd-l1* was synthesised as described previously (*Mann et al., 2014*). In situ hybridisation was performed at 68°C following the protocol as described previously (*Wu and Oh, 1996*).

## RNAscope

Gene-specific probes and the RNAscope Fluorescent Multiplex Reagent Kit (320850) were ordered form Advanced Cell Diagnostics. BP was collected from E8 to E10 chick embryos, fixed overnight in 4% PFA, and subsequently cryopreserved through a sucrose gradient (5%, 10%, 15%, 20%, and 30%). Samples were embedded in cryomolds using Tissue-Tek O.C.T compound and sectioned on a cryostat at 10–12 μm thickness. RNAscope hybridisation protocol was carried out based on the manufacturer's (ACD) suggestions. All fluorescent images were obtained on a Zeiss LSM900 confocal microscope.

## RNA-seq analysis

For bulk RNA-seq analysis, all genes with a $Log_2$ p-value >1 were considered significantly expressed in the distal BP region and all genes with a $Log_2$ <1 significantly expressed in the proximal BP region. Statistical significance levels were calculated by one-way ANOVA. For a gene to be considered 'differential', at least one region of the BP (proximal, middle, or distal) was required to be ≥0.5 RPKM. A fold change of ≥2 was imposed for the comparison between distal and proximal regions. A final requirement was that middle region samples had to exhibit RPKM values mid-way between proximal and distal regions to selectively capture transcripts with a gradient between the two ends. Bulk Affymetrix data were analysed for differentially expressed mRNAs encoding metabolic effector proteins that regulate cellular NADPH levels. Microarray signals were normalised using the RMA algorithm. The mRNAs expressed at significantly different levels in distal versus proximal BP were selected based on ANOVA analysis using the Partek Genomics Suite software package (Partek, St. Charles, MO, USA). **\*p < 0.05**. For detailed description of analysis and protocols please refer to *Mann et al., 2014*.

## Acknowledgements

The authors wish to thank Dan Jagger, Matthew Kelley, Jeremy Green, Michael Duchen, Gyorgy Szabadkai, and Thomas Coate for providing important comments and critical discussion on the manuscript. This project was supported by funds from the King's Prize Fellowship (King's College London) (ZFM), The Physiological Society (Physiological research Grant) (ZFM), and the Biotechnology and Biological Sciences Research Council (BBSRC) grant BB/V006371/1 (ZFM).

## Additional information

### Funding

| Funder | Grant reference number | Author |
| --- | --- | --- |
| Biotechnology and Biological Sciences Research Council | BB/V006371/1 | Zoe F Mann |
| Physiological Society | | Zoe F Mann |

| Funder | Grant reference number | Author |
|---|---|---|
| King's Prize Fellowship | | Zoe F Mann |

The funders had no role in study design, data collection, and interpretation, or the decision to submit the work for publication.

## Author contributions
James DB O'Sullivan, Data curation, Formal analysis, Investigation, Methodology, Writing – review and editing; Thomas S Blacker, Conceptualization, Methodology, Writing - original draft, Writing – review and editing; Claire Scott, Data curation, Formal analysis, Investigation, Methodology; Weise Chang, Mohi Ahmed, Data curation, Investigation, Methodology; Val Yianni, Formal analysis, Methodology; Zoe F Mann, Conceptualization, Resources, Data curation, Formal analysis, Supervision, Funding acquisition, Investigation, Visualization, Methodology, Writing - original draft, Project administration, Writing – review and editing

## Author ORCIDs
Thomas S Blacker  http://orcid.org/0000-0002-8949-6238
Val Yianni  http://orcid.org/0000-0001-9857-7577
Zoe F Mann  http://orcid.org/0000-0002-4916-9574

## Decision letter and Author response
Decision letter https://doi.org/10.7554/eLife.86233.sa1
Author response https://doi.org/10.7554/eLife.86233.sa2

## Additional files

### Supplementary files
• Source data 1. Two-way repeated measures analysis of variance (ANOVA) results for hair cell (HC) nuclear area. p-Values in the left-hand column indicate whether the proximal–distal differences in HC nuclear area was altered in inhibitor-treated explants compared to controls. The p-values in the interaction column indicate whether the proximal–distal differences in nuclear area changed after inhibitor treatment. YZ9 and Chdl1 + 2-deoxy-D-glucose (2-DOG) were the only treatments which did not produce a significant interaction, indicating preservation of the tonotopic gradient.

• Source data 2. Two-way repeated measures analysis of variance (ANOVA) results for hair cell (HC) luminal surface area (SA). p-Values in the left-hand column indicate whether the proximal–distal differences in HC luminal SA was altered in inhibitor-treated explants compared to controls. The p-values in the interaction column indicate whether the proximal–distal differences in luminal SA changed after treatment. YZ9 and Chdl1 + 2-deoxy-D-glucose (2-DOG) were the only treatments which did not produce a significant interaction, indicating preservation of the tonotopic gradient.

• MDAR checklist

### Data availability
All data and source data are available in manuscript and supporting files.

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
