## [Editor Report]

Morphogens such as Sonic hedgehog and Bone morphogenetic proteins (BMP) are known to establish the tonotopic organization of the cochlea. In this paper, the authors demonstrated for the first time that differential glucose metabolism along the basilar papilla (chicken cochlea) regulates the gradient of BMP7 signaling required for establishing tonotopy of the cochlea.

---

## [Decision Letter]

**Decision letter after peer review:**

[Editors’ note: the authors submitted for reconsideration following the decision after peer review. What follows is the decision letter after the first round of review.]

Thank you for submitting the paper "Gradients in cellular metabolism regulate development of tonotopy along the cochlea" for consideration by *eLife*. Your article has been reviewed by 3 peer reviewers, including Doris Wu as the Reviewing Editor and Reviewer #1, and the evaluation has been overseen by a Senior Editor. The following individuals involved in the review of your submission have agreed to reveal their identity: Jinwoong Bok (Reviewer #2); Katie Kindt (Reviewer #3).

Comments to the Authors:

We are sorry to say that, after consultation with the reviewers, we have decided that this work will not be considered further for publication by *eLife*.

Specifically, the causal relationship between the NADPH/NAD gradient and tonotopy has many outstanding issues that require further clarification. Strategies employed to inhibit cellular metabolism in this study could have many secondary effects that may not be related to the establishment of tonotopy. Furthermore, the mechanisms underlying the reported NADPH/NADH gradient in the developing BP and whether this gradient was truly affected by the experimental manipulations remain unclear.

*Reviewer #1 (Recommendations for the authors):*

This manuscript investigated the distribution of several metabolic indicators during basilar papilla (BP) development, which included the use of FLIM and live imaging of mitochondria. The authors showed a decreasing NADPH:NADH ratio from the proximal to basal region of the BP and provided evidence this differential ratio underlies the tonotopic organization of the organ. When glycolysis or more specifically the pentose phosphate pathway was blocked in vitro (presumably resulting in the reduction of NADPH production and NADPH:NADH ratio), hair cells in the proximal region of the BP adapted distal hair cell morphologies. Bmp7 expression, which is normally low, and Chordin expression, which is normally high in the proximal region of the developing papilla were concomitantly decreased when glycolysis was inhibited with 2-DOG treatments. Since metabolism could affect many cellular pathways, it is important to discern the specificity of blocking glycolysis in affecting tonotopy. First, is NADPH:NADH ratio indeed disrupted with 2-DOG treatments? Was FLIM conducted on specimens treated with 2-DOG? Could NADPH:NADH ratio be raised in the distal region and results in proximal hair cell morphologies? Tonotopy is a graded phenomenon along the cochlear duct but the proximal and apical regions of the BP but not the middle region were primarily compared in the study. The upregulation of Bmp7 and the downregulation of Chordin as a result of 2-DOG treatment did not seem graded but the control samples also did not show a clear gradient of expression of these genes. This could be confounded by the culture conditions as the graded expression pattern of Bmp7 and chordin in the normal BP published previously are clear. Therefore, cellular analyses of the middle region of the BP will help to substantiate the hypothesis.

More specific comments are listed as follows:

1) Lines 109-112. It is a bit puzzling that the proximal-distal gradient of NADPH/NADH ratio (tbound), which was proposed to be important for establishing the tonotopic gradient, was only observed at E6 and E14 but not at E9. What could be the explanation?

2) In Figure 5, blocking glycolysis resulted in hair cells of the proximal region adapting morphology of distal hair cells is an interesting result. However, is the proximal hair cell phenotype indeed caused by the disruption of tonotopy, or the distal hair cell pattern is the default and further differentiation into proximal hair cell morphologies require more glycolysis/NADPH? There seems to be a concomitant increase in hair cell number in the proximal region after 2-DOG treatment, which suggests other cellular events could be occurring and account for the hair cell morphological phenotype.

3) There is a lot of speculation without clear referencing on the role of developmental changes that were not found to show a gradient along the P-D axis e.g. Idh3a expression pattern going from apex to base gradient at E14 but switch to base to apex at E16, and LDHA expression switched between hair cells and supporting cells. It's a bit distracting and confusing to the main story.

4) In Figure 6, blocking the pentose phosphate pathway with 6-AN seems to affect the luminal size of distal HCs as well when compared to controls. In the distal region, there seems to be an increase in the density of the HCs. Thus, results in Figure 8 suppl 1 showing the lack of change in EdU uptake in the presence of 2-DOG is important to address this concern as well as to comments listed in #3. However, the images shown in Figure 8 suppl 1 were not convincing and seem to show there was an increase in proliferation in the proximal region of the treated samples when compared to control. Results in the distal region seemed variable. Providing quantification of these results would help to strengthen the conclusion that proliferation was not involved.

5) The lack of gradient differences in the mouse data, though interesting on its own, does not add to the manuscript.

Clarifications to improve the manuscript are listed as follows:

1. Figure 1E-H, Highlight the region in the low-power image where the inset was taken. Describe in the text the interpretations of the color differences in insets of G and H.

2. The term basilar papilla should be defined in the Introduction rather than just used in the title of Figure 3 and Materials and methods.

3. Line 345-346, cite reference indicating hair cell synapse maturation in BP at E16.

4. Figure 3B and C, it is not clear if the proximal to basal comparison of Aldh1a3 level was reversed between E6.5 and E14, or if the representation of the ratio was different between the two ages. I could have missed it, but I didn't see the description in the text in Figure 3C.

5. The differential expression of Ldha along the BP in Figure 3 was further investigated but not Aldh1a3, which was also differentially expressed at E6.5 and E14.

6. Figure 4E, F, it looks like 2NBDG uptake was variable among cells in both proximal and basal regions with some super bright labels in the proximal region. In any case, Figure 4E stated there was no difference in 2NBDG uptake between proximal and basal regions, but Figure 4F showed a difference at E14. What is the sample size for E14 in Figure 4F and the statistics applied? What is the take-home message here for readers?

7. Lines 391-393, there was no proximal to the basal gradient of NADPH/NADH at E9 as shown in Figure 1. So, it is hard to draw a conclusion about cytosolic versus mitochondria metabolism.

8. Line 540, would blocking PFK increase the PPP pathway?

9. Indicate which region of the cochlear duct was taken for images shown in Figure 9A-D.

10. Discussion (line 768-770). I don't think there is enough existing evidence to state that cell cycle exit in the BP progresses from distal to proximal regions. The data from Corwin's lab (cited reference) suggest a neural to the abneural spread of cell cycle exit, unlike their counterpart in the mouse inner ear.

*Reviewer #2 (Recommendations for the authors):*

This study investigated whether a proximal-to-distal gradient of glucose metabolism exists in the developing basilar papilla (BP) and whether the metabolic gradient plays a role in instructing the positional identity of auditory hair cells along the tonotopic axis. Several assays were applied to address the presence of metabolic gradient, including NAD(P)H FLIM, re-analysis of RNA-seq and microarray datasets, IDH3A immunofluorescence, and 2-NBDG/TMRM fluorescence. Using FLIM was an excellent choice as this method provides a spatial relationship between the metabolic state and tonotopic features of hair cells along the developing BP. To address the involvement of glucose metabolism in tonotopic patterning, several pharmacological interventions were applied to inhibit specific metabolic pathways, which revealed that cytosolic glycose metabolism, but not mitochondrial metabolism plays a role in tonotopic patterning. Furthermore, the signaling gradients known to direct tonotopic patterning are disrupted by inhibition of glucose metabolism. This study links morphogen gradients with metabolic states along the developing auditory organ, providing novel insights into how tonotopy patterning is regulated. However, several experimental results were inconsistent with each other in supporting the hypothesis that the NADPH/NADH gradient and glycolysis are directly involved in establishing the tonotopy in the BP. Furthermore, the mere measurement of the surface area of hair cells is not a sufficient criterion to determine tonotopy. Other features such as hair bundle height and the number of stereocilia per hair cell should be included.

Specific comments:

1) One of the major limitations of this study is that only one feature, the surface area, was used to investigate the tonotopy. Several tonotopic features of hair cell morphology have been reported, including hair cell surface area (used in this study), hair bundle height, stereocilia number, and hair cell length. It was shown that hair cell surface area is not much different between proximal and distal regions in the early stage, and as BP matures, the tonotopic differences in the surface area become obvious because the surface area in the proximal region is increased to a greater extent than the distal region (Tilney et al., 1986, Dev Biol). Thus, the smaller surface area in the proximal region of BPs treated with 2-DOG, SAM, or 6-AN compared to controls may be due to defective developmental progression induced by abnormal glucose metabolism. Therefore, tonotopic changes in hair morphology should be confirmed by analyzing other tonotopic features.

2) The author argued for an expression gradient of NADPH/NADH regulators along the tonotopic axis based on previously generated RNA-seq and microarray datasets (Nat Comm, 2014). However, while LDHA mRNA levels are ~4-fold higher in the distal region than in the proximal region in the E6.5 RNA-seq data (Figure 3B), LDHA fluorescent intensity does not differ significantly between the proximal and distal regions at E7 (Figure 4 Supplement 2B). Therefore, it is essential to validate the expression gradients of selected genes from RNA-seq and microarray datasets using more reliable quantitative methods such as qRT-PCR or Western blot.

3) Based on the 2-NBDG and TMRM fluorescence results (Figure 4), the authors concluded that the gradient of NADPH/NADH originates specifically from tonotopic differences in cytosolic metabolism (line 392). However, there is a temporal gap between the two gradients. Tonotopic differences in NADPH/NADH are already significant at E6 (Figure 1), whereas the tonotopic difference in cytosolic glucose uptake is not significant until E14 (Figure 4). This temporal discrepancy makes it difficult to conclude that NADPH/NADH gradient (E6) originates from the glucose uptake gradient (E14). Similarly, it is difficult to link a causal relationship of tonotopic differences between surface area and glucose uptake because while glucose uptake is not significantly different until E14 (Figure 4), the surface area can be changed by blocking glycolysis at E6-E8 (Figure 5). To argue that metabolic gradients play a role in tonotopic patterning, the tonotopic gradient of NADPH/NADH, its regulators, and glucose uptake should be clearly correlated spatially as well as temporally in the developing BP.

4) Changes in Bmp7 and Chdl1 gradients by 2-DOG treatment (Figure 8) provide a molecular mechanism by which glucose metabolism regulates the tonotopic patterning of BP. However, 2-DOG has been shown to influence several cellular processes other than glycolysis, such as angiogenesis and autophagy. Therefore, it is crucial to confirm that the changes in Bmp7 and Chdl1 gradients by 2-DOG are due to inhibition of glucose metabolism but not due to other cellular effects. This will be confirmed if SAM and 6-AN, but not YZ9 and UK5099, which induced surface area changes similar to 2-DOG, also cause similar changes in Bmp7 and Chdl1 gradients.

1) Specify the model system in the abstract since birds and mammals appear to have different mechanisms for tonotopic patterning.

2) Figure 1I., τbound values were consistently higher in the proximal than distal regions from E6 to E16, although statistical analysis indicates significant differences at E6 and E14 but not at E9 and E16. Non-significance at E9 and E16 may be due to the small number of n, which made statistical power not strong enough to draw statistical significance. To give the readers a better idea of how variable τbound values are during development, it is recommended that the graphs on the left are presented as a boxplot with individual data points. Also, provide error bars and statistical analysis for the graphs on the right.

3) Figure 2C and D. Similar to the point above, it is recommended to present the graphs as a box plot with individual data points. Does "n = 10 biological replicates" mean 10 different BP samples? If so, indicate how many hair cells and supporting cells per BP sample were used to calculate τbound values?

4) Line 323. There are several instances where the authors refer to "other systems". It will be better to specify the system the authors refer to so that the readers get an idea of how the "other systems" are related to the auditory system. e.g., "studies in other systems" in lines 458, 505, and 685.

5) Figure 8. Why are the surface areas of control BPs so different between B (~40 μm2 in proximal) and C (~20 μm2 in proximal)?

6) Figure 8 Supplement 1. It is difficult to conclude based on the images whether 2-DOG treatment affects proliferation in the developing BP. Counting EdU+ cells will help to conclude whether or not cell proliferation is changed by inhibiting glucose metabolism.

7) Figure 9. The results from the mouse organ of Corti are interesting but do not appear to be relevant to the rest of the manuscript, since differences in τbound values were compared only between HCs and SCs, but not along the tonotopic axis. It is recommended to either present the τbound values between the proximal and distal regions of the organ of Corti or save them for further study.

8) Line 853. While the Shh gradient is present along the auditory organ in both birds and mammals, the Bmp7 gradient is present only in birds but not in mammals (Son et al., 2014, PNAS).

9) It is clear that inhibition of glucose metabolism by 2-DOG, SAM, and 6-AN induces the proximal BPs to become more like distal BPs, such as decreased surface area and increased Bmp7 expression in the proximal region. Is it possible to conduct the opposite experiment, i.e., activation of glucose metabolism, which may cause distal BP to become like proximal BP?

*Reviewer #3 (Recommendations for the authors):*

In this manuscript, Blacker et al. examine cellular metabolism primarily in the chick auditory organ, the basilar papilla (BP). They use FLIM to visualize NAD(P)H in live tissue to study changes in glucose metabolism within the auditory epithelium over the course of development. They show that there are differences in metabolism between HCs and SCs. They also show that there are spatial differences in metabolism along the length of the BP at specific developmental time points. The authors bolster evidence for this metabolic gradient using existing RNAseq datasets that show differential expression of metabolic genes along the proximal-to-distal axis of the BP. They then use pharmacology to manipulate glucose metabolism (glycolysis, pentose phosphate pathway, and the TCA cycle). The authors show that altering glucose metabolism early in glycolysis or at the level of the pentose phosphate pathway results in a loss of tonotopy in the BP. In contrast, inhibiting glucose metabolism at the level of the TCA cycle impaired HC differentiation. Further, they show that inhibiting glycolysis alters morphogen gradients that are required for the emergence of tonotopy within the developing BP. Overall this work suggests that specific metabolic pathways may drive developmental changes in morphogen gradients and – perhaps – contribute to HC differentiation. This is an exciting advance for the hearing field, especially with respect to potential regenerative therapies.

Strengths of the paper:

1. The use of FLIM to study NAD(P)H fluorescence in living HCs and SCs across the auditory epithelium during the course of development.

2. The powerful use of pharmacology to probe the role of glucose metabolism in tonotopy in long-term cultures of BP explants.

3. Using multiple approaches for the study: live NAD(P)H imaging, RNAseq data, pharmacology, vital dyes, and immunohistochemistry.

4. Linking distinct metabolic pathways to developmental events is a novel and exciting area of research.

Weaknesses of paper:

1. The majority of the manuscript uses t-tests for comparisons. In many cases, a 2-way ANOVA is needed for comparisons. It is not clear that all the findings will hold up to more rigorous statistical analysis.

2. Substantial work is needed to enhance the clarity of the manuscript. There is a limited introduction to the different facets of glucose metabolism highlighted in this paper. Metabolism is a complex topic that a broad audience will not be intimately familiar with. In addition, the paper is rich with data. The authors discuss differences in metabolism along 3 variables (space, time, and cell type). Occasionally it is unclear what type of difference is being discussed and the trend of the difference.

3. Manipulations inhibiting glycolysis and oxidative phosphorylation over extended periods of time could have many secondary effects on development related to a lack of energy supply.

Overall, this is a very interesting line of research. Currently, the manuscript needs revision prior to publication.

1. Statistics. The majority of the manuscript uses t-tests, which are not appropriate when making multiple comparisons (ie: proximal vs. distal over multiple stages of development). Currently, it is not clear that all the findings will hold up to more rigorous statistical analysis. For example, Figure 1I, 1J, 2C, 2D, 4D, 4F, 4G, 4I, 4J, all of Figure 4 Supp. 1, Figure 4 Supp. 2B, 5C, Figure 5 Supp. 1C, 6D, Figure 6 Supp. 1B, 7C, 8B, 8C, 9E, and 9F should be analyzed using two-way ANOVA. It is not statistically sound to pick only specific comparisons to make.

Additionally, in some cases, there are stars above bars with no indication of what condition is being compared to (ie. Figure 9E). Figure 3A lacks statistical analysis altogether. Finally, there are several cases where the authors say that there is a "significant" difference but do not show that it is statistically significant (for example, the caption of Figure 4I says there is a significant decrease between E9 and E14, but there is no indication of statistical significance in the figure). Differences that are not statistically significant can be discussed as trends, but should not be called significant. Similarly, differences that are statistically significant should not be discussed as "no difference" (ex. Figure 4E/F caption but Figure 4F shows a statistically significant difference; Figure 4 Supp. 2).

2. Enhance the clarity of the manuscript. The manuscript is quite well written but currently, there is limited text in the introduction outlining the relevant parts of glucose metabolism highlighted in this paper. Or how NADPH and NADH feed into these pathways. In addition, the authors discuss differences in metabolism along 3 variables (space, time, and cell type), and occasionally it is not clear what type of difference is being discussed – what is increased or decreased and where. Schematics like those in 2E are helpful to the reader.

3. Manipulations inhibiting glycolysis and oxidative phosphorylation over extended periods of time could have many secondary effects on development. The authors should lighten claims of causation. At a minimum, there should be a discussion of possible secondary effects and the need for more targeted manipulations to draw a convincing causative link between cellular metabolism and the development of tonotopy within the BP.

4. The authors detect a proximal-to-distal gradient in tbound at E6 that "was still present at E14" and discuss this as a gradient "along the tonotopic axis between E6 and E14". However, their figure does not show a significant difference between the proximal and distal regions at E9. This could be related to a lack of statistical power (the n is lower for E9), but could alternatively represent a real fluctuation in the metabolic state over time. The authors should include a discussion of their E9 data, rather than glossing over it.

5. Although it may be beyond the scope of this paper, it would be nice to come full circle back to NAD(P)H imaging. Do the alteration to the PPP pathway lead to a loss of the NADPH/NADH gradient along the proximal-distal axis in the developing BP?

6. The authors make several claims regarding differences in TMRM. They should also consider that this cationic vital dye may enter HCs via mechanotransduction channels. This form of entry may explain why labeling is stronger in older HCs compared to SCs or younger HCs.

[Editors’ note: further revisions were suggested prior to acceptance, as described below.]

Thank you for resubmitting your work entitled "Crosstalk between glucose metabolism and Bmp7-Chordin-like 1 signalling specifies tonotopic identity in developing hair cells." for further consideration by *eLife*. Your revised article has been evaluated by Kathryn Cheah (Senior Editor) and a Reviewing Editor.

The manuscript has been improved but there are some remaining issues that need to be addressed, as outlined below:

Essential revisions:

While all three reviewers felt that the revised manuscript is a substantial improvement there are still some outstanding issues:

1) The Chordin rescue experiment requires additional controls and clarification.

2) Additional inconsistencies and quantification were requested by all three reviewers.

3) Comparing treatments of SAM or 6-AN to 2-DOG in the ability to affect Bmp7 signal, for example.

*Reviewer #1 (Recommendations for the authors):*

This revised manuscript is much improved and streamlined. The authors provided additional results showing that Chordin like-1, which is normally expressed in the proximal region of the BP could rescue the distalizing effect of 2-DOG in the proximal region, thus strengthening the link between metabolism and tonotopy. However, in this type of rescue experiment, 2-DOG and Chordin like-1 (Chdl-1) treatments should be included as controls within the same experiment, rather than using controls with no treatment alone. While readers could extrapolate 2-DOG effects from other figures, there was no description of Chdl-1 in dosage, condition, or results in the revised manuscript. Furthermore, the statement in lines 800-802 and results in Supplementary Table 1 seem to suggest that Chdl-1 treatment alone does not affect the tonotopic gradient by itself. Is that true? If so, this result is not consistent with previous studies.

Editorial suggestions to help improve the manuscript:

1) Line 28, what does flattening of hair cell morphology mean? Do you mean shorter hair bundles in the proximal versus the distal BP? If so, the images in this study are not clear enough for naïve readers to draw that conclusion.

2) Line 54, what does inner sense organs means? Are you referring to the inner ear or tissues deep in the body?

3) Figure 1 is very helpful but could use some revision: a) consider putting labels for the compartments at the top of the figure with a larger font. It may help readers to see the longitudinal lines represent mitochondrial membrane, which is not apparent, b) increase the font size of the pathways, c) no explanation for labels 1, 2, and 3 in the legend and I am not sure if they meant to illustrate something more than the color scheme, and d) add PKM2 to the metabolic flowchart.

4) Lines 293 and 295, a bit odd to split PKM2 results into two separate paragraphs.

5) Figure 2 a) Consider combining panels D and K, and expand B and C. Hard to read the label in C, b) I am still not clear why αbound level (dotted purple line) is in a steady state based on the purple and green curve and there is no explanation of αbound in the text, c) Panels E-H could be slightly bigger, d) move Scale bars=50um to the end of description for (E-H).

6) Line 205, probably better just to state τbound gradient not related to glucose uptake?

7) Line 236, the sentence is a bit awkward.

8) Line 292, define PKM2 when it first appeared in the text.

9) Line 339, Figure 3 or Figure 5?

10) Line 342, Figure without a number.

11) Lines 340-343, State in the text lower pHrodo Red signal meant higher pH.

12) No quantification of Edu-labeled cells in Figure 6 suppl 2.

13) Line 484, does SAM inhibit both glycolysis and PPP pathways similar to 2-DOG?

14) Typo in the last sentence of the legend for Figure 7 supplement 1.

15) Incomplete last sentence found in the legend for Figure 8.

16) Figure 9. The images in E are better in this revision. However, there is some inconsistency between the text and the figure legend. In the text, explants were established at E6.5 and incubated for 72 hours. In the legend, it was stated the in vitro incubation was for 7 days.

17) Figure 9. It may be better to put the number and statistics in the panels that they belong rather than at the end of the figure. Sample numbers need to be provided for panel E as well.

18) Supplemental Table 1. Are all p values in the left-hand columns controls? Confused by the statement in line 799, "control or inhibitor treatment groups". Or it should be controlled for inhibitor treatment groups? If so, the p values in the right-hand column are results from individual treatments. Then, Chdl-1 treatment did not affect the tonotopic differences observed in controls?

*Reviewer #2 (Recommendations for the authors):*

Figure 7 supplementary 2.

Based on the statistical analysis, the authors concluded that YZ9 treatment had no effect on HC morphology along the tonotopic axis. However, the graph in (B) shows that the luminal area of hair cells was dramatically reduced in the proximal but not in the distal region by YZ9 treatment, resulting in no significant difference between the proximal and distal regions. This suggests a loss of tonotopic identity of HC morphology by YZ9 treatment, similar to 2-DOG, SAM, or 6-AN. To support the authors' argument, it is important to confirm that YZ9 treatment did not induce tonotopic changes in HC morphology. Double-checking with other tonotopic parameters, such as HC nuclear area and cell number, will help to clarify this issue.

Figure 9.

(E) The Bmp7 expression domain in the control explant is peculiar in that it has a lining around the distal region, unlike the control image in Figure 9 supplement 1. Figure 3 supplement 4 shows that Bmp7 is normally expressed in the sensory domain.

Bmp7 in situ signals are generally stronger in 2-DOG-treated BP than in control, but there still seems to be a proximal-distal gradient with a generally stronger in situ signal. qRT-PCR with each half of the BPs, as done in their previous publication, will clearly show if the gradient is disrupted.

Figure 9 supplement 1.

The previous review questioned whether the changes in tonotopic identity caused by 2-DOG were due to the inhibition of glucose metabolism rather than other cellular effects. This question was crucial to support the authors' conclusion, as 2-DOG has been shown to affect several cellular processes other than glycolysis. The authors showed that blocking mitochondrial metabolism did not alter the Bmp7 gradient, but this doesn't solve the problem of the specificity of 2-DOG. It is worth investigating whether SAM and 6-AN, which induce luminal area changes similar to 2-DOG, also disrupt the same molecular pathways, such as the Bmp7 and Chld1 gradients.

*Reviewer #3 (Recommendations for the authors):*

This manuscript is substantially improved compared to the previous version. Thank you for all the hard work you put into responding to reviewer concerns. A revised manuscript would be welcomed by this reviewer.

– The RNAseq and microarray analyses are good for hypothesis generation, but the differential expression the authors detect often does not match the immunohistochemistry and RNA scope data they provide. It is especially confusing to say that there is no difference in LDHA and IDH3A protein expression across the BP, then suggest that differential RNA expression of these genes could drive the metabolic gradient the authors observe. Some of the variability in the data may be related to differences in animal age across experiments, but this variability makes it a bit challenging to interpret the data. In addition, not all the genes listed in Figure 3 Supp. 4A seems to reach statistical significance. The authors should consider framing Figure 3 Supp. 4A as a way to generate hypotheses that they then test with additional experiments, rather than claiming that they identified a whole set of differentially expressed genes when in fact they present contradictory data within the same manuscript. They should also clarify how they decided which metabolic RNAs to examine further.

– Although the authors used BMP7 as a positive control for RNA scope, there does not appear to be a proximal-to-distal difference in BMP7 fluorescence. In addition, the authors do not provide any negative controls to show a lack of fluorescence in the absence of gene-specific RNA scope probes. Without successful controls, the experimental data is hard to interpret. The authors should consider providing better controls.

– Line 297-298: "By controlling a metabolic feedback loop that regulates the switch between glycolysis and OXPHOS…" The authors previously showed data suggesting that differences in the NAD(P)H/NADH ratio are not due to changes in mitochondrial OXPHOS. Here, though, they suggest that PKM2 controls the amount of OXPHOS and is differentially expressed across the BP. How can we reconcile these two points?

– Figure 4, 5: How was the quantification in 4B-D and 5B performed? Are fluorescence values provided per HC/per SC? As HC size varies across the proximal-distal axis of the BP, it is not fair to quantify raw total fluorescence. The fluorescence should be normalized by the area occupied by each cell type.

– Figure 5: Could proximal-to-distal differences in pHrodo Red fluorescence result from the gradient of HC differentiation (i.e. more mature distal HCs, less mature proximal HCs) across the tonotopic axis at E9? In other words, do young HCs have a higher pHi that decreases as hair cells mature? This could explain the proximal-to-distal differences shown in Figure 5.

– Figure 6 Supp. 2 – To claim that there is no increased proliferation in response to 2-DOG, EdU+ cells should be quantified in control vs. 2-DOG cultures. It seems that the authors have only provided general cell counts (in C) and not EdU+ cell counts.

– Do the authors have any evidence that the various drug treatments are not simply blocking cell growth and differentiation? As mentioned by another reviewer, the distal-like phenotype could be a "default"; perhaps proximal HCs simply aren't maturing. It would be nice to show examples of cultures at 1DIV vs. 7 DIV to demonstrate how much cells grow during this time and how treated cultures at 7 DIV compare to immature 1 DIV cultures. (SAM treatment especially looks like it might impact hair bundle morphology in the proximal BP.)

– Figures 7 Supp. 2 and Figure 8 continue to rely on a single morphological measure of HC morphology as a readout of tonotopy. The authors should add additional quantification (cell density, nuclear area…), as has been provided for other manipulations.

– Figure 9 and Figure 9 Supp. 1: In the absence of quantification of the in situ hybridization data, please state how many examples show similar Bmp7 and Chdl1 gradients.

– To bolster the claim that 2-DOG treatment is specifically disrupting tonotopy and not just the differentiation of proximal HCs (as pointed out by Reviewer 1), the authors should quantify HC density in Figure 10. If HC density is also restored by providing Chdl-1 in the presence of 2-DOG, it will bolster the hypothesis that altered glucose metabolism has a specific effect on HC positional identity.

[Editors’ note: further revisions were suggested prior to acceptance, as described below.]

Thank you for resubmitting your work entitled "Crosstalk between glucose metabolism and morphogen signalling specifies tonotopic identity in developing hair cells" for further consideration by *eLife*. Your revised article has been evaluated by Kathryn Cheah (Senior Editor) and a Reviewing Editor.

The manuscript has been improved but there are some remaining editorial issues that need to be addressed, as outlined below:*Reviewer #1:*

The results of the revised manuscript are coming together nicely. To appropriately demonstrate the rescue effect of Chd1 on 2DOG requires the control, 2-DOG, and 2-DOG+Chd1 be all conducted within the same experiment. Although such an experiment was not provided, the authors have made a good faith effort in demonstrating a dose-response effect of chordin in reducing the differential differences between proximal and distal BP, and the revision is acceptable. Additionally, I have some editorial suggestions. These are merely suggestions, and the acceptance of the manuscript is not dependent on making these changes.

1) The experimental evidence suggests that the tonotopic morphogens are regulated by glucose metabolism. I don't see evidence showing that these morphogens regulate metabolism in return. Therefore, I question the choice of using the phrase "Cross talk" in the title.

2) The abstract, though clear, could be more concise and informative. I feel like it is important to summarize this 32-figure manuscript well!

3) Include PKm2 in the schematic diagram of Figure 1.

4) The first part of the Discussion reads more like an introduction.

*Reviewer #2:*

The revised manuscript is much improved with additional experimental data that strongly support the authors' conclusions. I appreciate the authors' efforts to make the manuscript in much better shape.

I have just one question. From the diagrams in Figure 7-supplement 2B and Figure 8B, it appears that PFK and PKM2 act in the same glycolysis pathway. However, inhibition of either resulted in different phenotypes. While inhibition of PFK by YZ9 did not affect tonotopic patterning, inhibition of PKM2 by shikonin disrupted tonotopic patterning, albeit at the epithelial level. Why did two inhibitors of the same pathway have different effects on tonotopic patterning?*Reviewer #3:*

The manuscript is greatly improved and nearly ready for publication.

There are a few typos and points to clarify that I would recommend to the authors. These recommendations should not require additional experiments.

List of recommendations for clarity:

Figure 2 Supplement 1 – this Figure is really helpful and we would recommend making it part of a main figure (perhaps you can condense Figure 2G, as these two plots show the same data, in favor of this helpful schematic). The clarity of the FLIM section is massively improved since the first version of the paper. The one link that is still hard to make is how the values of Tbound inform flux through the PPP vs. glycolysis. This figure beautifully illustrates exactly that. Also, consider adding PPP vs. glycolysis to the gradient in the Figure 2H schematic, and/or moving the sentence in line 109 to line 101 in the main text. Although all of this knowledge is obvious to those who think about metabolism all the time – these concepts can be confusing for someone not in the field.

We would recommend adding Pkm2 to Figure 1, as there is an entire figure focused on Pkm2.

Figure 3 Supp. 3E. Are you detecting statistically significant proximal-to-distal changes in Bmp7 here? It is not indicated in the quantification (no stars or "ns"). If the positive control is not showing a statistically significant change, it is hard to interpret the other data.

---

## [Author Response]

[Editors’ note: the authors resubmitted a revised version of the paper for consideration. What follows is the authors’ response to the first round of review.]

Comments to the Authors:We are sorry to say that, after consultation with the reviewers, we have decided that this work will not be considered further for publication by eLife.Specifically, the causal relationship between the NADPH/NAD gradient and tonotopy has many outstanding issues that require further clarification. Strategies employed to inhibit cellular metabolism in this study could have many secondary effects that may not be related to the establishment of tonotopy. Furthermore, the mechanisms underlying the reported NADPH/NADH gradient in the developing BP and whether this gradient was truly affected by the experimental manipulations remain unclear.

In attempt to further characterise a role for metabolism in tonotopic patterning, we have included additional experiments showing a rescue of tonotopic patterning when explants are treated with medium containing both 2-DOG and Chordin like-1. We believe the observed effects when treating with 2-DOG and Chordin like-1 together (new data Figure 10) support a role for metabolism specifying tonotopy in the developing BP.

Reviewer #1 (Recommendations for the authors):This manuscript investigated the distribution of several metabolic indicators during basilar papilla (BP) development, which included the use of FLIM and live imaging of mitochondria. The authors showed a decreasing NADPH:NADH ratio from the proximal to basal region of the BP and provided evidence this differential ratio underlies the tonotopic organization of the organ. When glycolysis or more specifically the pentose phosphate pathway was blocked in vitro (presumably resulting in the reduction of NADPH production and NADPH:NADH ratio), hair cells in the proximal region of the BP adapted distal hair cell morphologies. Bmp7 expression, which is normally low, and Chordin expression, which is normally high in the proximal region of the developing papilla were concomitantly decreased when glycolysis was inhibited with 2-DOG treatments. Since metabolism could affect many cellular pathways, it is important to discern the specificity of blocking glycolysis in affecting tonotopy.

To address this issue, we have used multiple inhibitors that target different branches of glucose flux. The same phenotype was produced in response to all inhibitors except YZ9. From a comparison of the statistical results between all inhibitor experiments (Supplementary table 1), we believe that the observed changes in cell morphology are a result of dysregulated metabolism rather than off-target effects.

First, is NADPH:NADH ratio indeed disrupted with 2-DOG treatments? Was FLIM conducted on specimens treated with 2-DOG?

For FLIM measurements, treated explants need to be transferred from culture dishes to FLIM imaging dishes. This causes additional mechanical and osmotic stress, which may impact metabolism and introduce variation to NAD(P)H FLIM measurements. In addition, there are gross morphological changes during long-term culture, and given that fluorescent markers cannot be used in combination with FLIM it is not feasible to identify proximal and distal ends.

Could NADPH:NADH ratio be raised in the distal region and results in proximal hair cell morphologies?

The reviewers raise an excellent point, and whilst this would be a valuable experiment, we believe it is beyond the scope of the current study. There are no pharmacological agents available to specifically target the NADPH/NADH ratio in vitro. This could potentially be done using an in ovo electroporation approach to knock out NAD kinase in different regions of the BP. However, as it is not possible to control the region-specific uptake of constructs during in ovo electroporation, it would not be possible to target NADPH/NADH pools in proximal and distal BP regions specifically.

Tonotopy is a graded phenomenon along the cochlear duct but the proximal and apical regions of the BP but not the middle region were primarily compared in the study. The upregulation of Bmp7 and the downregulation of Chordin as a result of 2-DOG treatment did not seem graded but the control samples also did not show a clear gradient of expression of these genes. This could be confounded by the culture conditions as the graded expression pattern of Bmp7 and chordin in the normal BP published previously are clear. Therefore, cellular analyses of the middle region of the BP will help to substantiate the hypothesis.

New images from repeated in situs have been added for clarity in Figure 9 were re-ran to more clearly show the change in gradient following treatment. See Figure 9 panel E. In addition, we have included in situ data for Bmp7 expression following treatment with the OXPHOS inhibitor UK5099 (Figure 9 Supplement 1).

More specific comments are listed as follows:1) Lines 109-112. It is a bit puzzling that the proximal-distal gradient of NADPH/NADH ratio (tbound), which was proposed to be important for establishing the tonotopic gradient, was only observed at E6 and E14 but not at E9. What could be the explanation?

Additional n numbers have been added to this data set, and all data have been re-analysed for statistical significance using 2-way ANOVA testing. Following new analysis, the gradient in tbound (NADPH/NADH) is significant at E6, E9 and E14 (see revised Figure 2).

2) In Figure 5, blocking glycolysis resulted in hair cells of the proximal region adapting morphology of distal hair cells is an interesting result. However, is the proximal hair cell phenotype indeed caused by the disruption of tonotopy, or the distal hair cell pattern is the default and further differentiation into proximal hair cell morphologies require more glycolysis/NADPH?

This is interesting point! The distal pattern may indeed be the default. It may also be the altered PPP metabolism interferes with normal cell cycle progression and differentiation. Cell cycle exit and cell size, growth are tightly coupled with metabolism, in particular PPP metabolism and pH (see new data added in Figure 4). Altering these gradients along the BP during development with 2-DOG, 6AN or SAM may therefore interfere with the normal patterning in cell size and number. We have included this argument as a key point in the discussion (Line 813).

There seems to be a concomitant increase in hair cell number in the proximal region after 2-DOG treatment, which suggests other cellular events could be occurring and account for the hair cell morphological phenotype.

We have re-done the EdU experiments and quantified cell density in the proximal and distal regions in control and treated explants. Overall blocking glycolysis consistently reduces EdU staining. Cultures were set up at E8 and maintained in culture for 48h covering the active phase of mitotic cell division. The increase in cell density in the proximal but not distal region is consistent to what we saw with Bmp7 and based on new data (Figure 6 Supplement 2), does not seem to be driven by an increase in proliferation.

3) There is a lot of speculation without clear referencing on the role of developmental changes that were not found to show a gradient along the P-D axis e.g. Idh3a expression pattern going from apex to base gradient at E14 but switch to base to apex at E16, and LDHA expression switched between hair cells and supporting cells. It's a bit distracting and confusing to the main story.

We thank the reviewer for this comment and fully agree that the previous way in which the data were presented was unclear. This section of the manuscript has now been significantly re-written for clarity and new results have been included to support our overarching hypothesis (Figures 4 and 5).

We refer the reviewer to Line 206 section beginning “Expression of Lactate and isocitrate dehydrogenases in the developing BP…”.

4) In Figure 6, blocking the pentose phosphate pathway with 6-AN seems to affect the luminal size of distal HCs as well when compared to controls. In the distal region, there seems to be an increase in the density of the HCs. Thus, results in Figure 8 suppl 1 showing the lack of change in EdU uptake in the presence of 2-DOG is important to address this concern as well as to comments listed in #3. However, the images shown in Figure 8 suppl 1 were not convincing and seem to show there was an increase in proliferation in the proximal region of the treated samples when compared to control. Results in the distal region seemed variable. Providing quantification of these results would help to strengthen the conclusion that proliferation was not involved.

We refer the reviewer to new data included in Figure 6 Supplement 2 and quantification for changes in proliferation and cell density.

We have also included further analysis for HC nuclear area in addition to luminal surface area measurements. Higher magnification images have also been included in the Supplementary data to highlight changes in HC luminal surface areas (Figure 6 Supplement 1 and Figure 7 Supplement 1).

5) The lack of gradient differences in the mouse data, though interesting on its own, does not add to the manuscript.

We agree that these data distract from the main focus of the study and have therefore removed it from the manuscript.

Clarifications to improve the manuscript are listed as follows:1. Figure 1E-H, Highlight the region in the low-power image where the inset was taken. Describe in the text the interpretations of the color differences in insets of G and H.

This Figure has now been significantly revised (see new Figure 2).

2. The term basilar papilla should be defined in the Introduction rather than just used in the title of Figure 3 and Materials and methods.

This definition has been included in both the abstract and introduction.

3. Line 345-346, cite reference indicating hair cell synapse maturation in BP at E16.

This discussion point is no longer included in the manuscript.

4. Figure 3B and C, it is not clear if the proximal to basal comparison of Aldh1a3 level was reversed between E6.5 and E14, or if the representation of the ratio was different between the two ages. I could have missed it, but I didn't see the description in the text in Figure 3C.

Aldh3 expression data has been removed. See updated Figure 3 Supplement 4.

5. The differential expression of Ldha along the BP in Figure 3 was further investigated but not Aldh1a3, which was also differentially expressed at E6.5 and E14.

This section of the paper has now been significantly revised and Aldh3 expression data are no longer included.

6. Figure 4E, F, it looks like 2NBDG uptake was variable among cells in both proximal and basal regions with some super bright labels in the proximal region. In any case, Figure 4E stated there was no difference in 2NBDG uptake between proximal and basal regions, but Figure 4F showed a difference at E14. What is the sample size for E14 in Figure 4F and the statistics applied? What is the take-home message here for readers?

This is now addressed in the Results section (line 200 and Figure 3 Supplement 2). Sample size has been added to the legend for Figure 3 Supplement 2, n = 5.

To provide further clarity and hopefully address the reviewer’s concern, 2-NBDG provides an indirect measure of intracellular glucose metabolism. The dye provides a measure of glucose uptake across the plasma membrane, but subsequent to its phosphorylation by hexokinase (HK) 2-NBDG does not provide a measure of whether glucose is catabolised in glycolysis, or via the PPP. It is therefore not clear how the 2-NBDG signal should be interpreted relative to the FLIM gradient in _bound_. Further studies are needed to untangle the how the two assays and their combined signals are best interpreted as read-outs of metabolic phenotype.

7. Lines 391-393, there was no proximal to the basal gradient of NADPH/NADH at E9 as shown in Figure 1. So, it is hard to draw a conclusion about cytosolic versus mitochondria metabolism.

Upon increasing the sample size (n = 4) the gradient in _bound_ is now significant at both E6 and E9. See revised Figure 2.

8. Line 540, would blocking PFK increase the PPP pathway?

Blocking PFK should not affect flux into the PPP, as this step is glucose catabolism is regulated by activity of G6PDH. The PPP can however feed metabolites back into the main branch of glycolysis from the non-oxidative branch, so may contribute to activity in the lower branch of glycolysis.

9. Indicate which region of the cochlear duct was taken for images shown in Figure 9A-D.

Added to Figure 9.

10. Discussion (line 768-770). I don't think there is enough existing evidence to state that cell cycle exit in the BP progresses from distal to proximal regions. The data from Corwin's lab (cited reference) suggest a neural to the abneural spread of cell cycle exit, unlike their counterpart in the mouse inner ear.

This section has now been re-written: See paragraph beginning line 814.

Reviewer #2 (Recommendations for the authors):This study investigated whether a proximal-to-distal gradient of glucose metabolism exists in the developing basilar papilla (BP) and whether the metabolic gradient plays a role in instructing the positional identity of auditory hair cells along the tonotopic axis. Several assays were applied to address the presence of metabolic gradient, including NAD(P)H FLIM, re-analysis of RNA-seq and microarray datasets, IDH3A immunofluorescence, and 2-NBDG/TMRM fluorescence. Using FLIM was an excellent choice as this method provides a spatial relationship between the metabolic state and tonotopic features of hair cells along the developing BP. To address the involvement of glucose metabolism in tonotopic patterning, several pharmacological interventions were applied to inhibit specific metabolic pathways, which revealed that cytosolic glycose metabolism, but not mitochondrial metabolism plays a role in tonotopic patterning. Furthermore, the signaling gradients known to direct tonotopic patterning are disrupted by inhibition of glucose metabolism. This study links morphogen gradients with metabolic states along the developing auditory organ, providing novel insights into how tonotopy patterning is regulated. However, several experimental results were inconsistent with each other in supporting the hypothesis that the NADPH/NADH gradient and glycolysis are directly involved in establishing the tonotopy in the BP. Furthermore, the mere measurement of the surface area of hair cells is not a sufficient criterion to determine tonotopy. Other features such as hair bundle height and the number of stereocilia per hair cell should be included.

It is very challenging to quantify bundle morphology accurately in long-term explants (cultured for more than 7 days). We believe accurate bundle measurements would need to be done using an in ovo electroporation model.

Specific comments:1) One of the major limitations of this study is that only one feature, the surface area, was used to investigate the tonotopy. Several tonotopic features of hair cell morphology have been reported, including hair cell surface area (used in this study), hair bundle height, stereocilia number, and hair cell length. It was shown that hair cell surface area is not much different between proximal and distal regions in the early stage, and as BP matures, the tonotopic differences in the surface area become obvious because the surface area in the proximal region is increased to a greater extent than the distal region (Tilney et al., 1986, Dev Biol). Thus, the smaller surface area in the proximal region of BPs treated with 2-DOG, SAM, or 6-AN compared to controls may be due to defective developmental progression induced by abnormal glucose metabolism. Therefore, tonotopic changes in hair morphology should be confirmed by analyzing other tonotopic features.

Additional analysis of HC nuclear area and cell density in proximal and distal regions have been included.

2) The author argued for an expression gradient of NADPH/NADH regulators along the tonotopic axis based on previously generated RNA-seq and microarray datasets (Nat Comm, 2014). However, while LDHA mRNA levels are ~4-fold higher in the distal region than in the proximal region in the E6.5 RNA-seq data (Figure 3B), LDHA fluorescent intensity does not differ significantly between the proximal and distal regions at E7 (Figure 4 Supplement 2B). Therefore, it is essential to validate the expression gradients of selected genes from RNA-seq and microarray datasets using more reliable quantitative methods such as qRT-PCR or Western blot.

We thank the reviewer for raising this important point. For this specific study, we chose to use RNA-scope or immunohistochemistry to look at expression of metabolic enzymes along the BP as these techniques preserve the spatial resolution needed to look at metabolism on a cell-by-cell basis. Using western blot and qPCR analysis, whilst informative, would not maintain the spatial and single cell resolution within the tissue. A further caveat is that metabolic reprograming is challenging to detect at the transcriptional level, as it is regulated at the protein level.

We have included additional analysis for GOT2, LDHB and PKM2 (see Figure 3 Supplement 4 and Figure 4).

3) Based on the 2-NBDG and TMRM fluorescence results (Figure 4), the authors concluded that the gradient of NADPH/NADH originates specifically from tonotopic differences in cytosolic metabolism (line 392). However, there is a temporal gap between the two gradients. Tonotopic differences in NADPH/NADH are already significant at E6 (Figure 1), whereas the tonotopic difference in cytosolic glucose uptake is not significant until E14 (Figure 4). This temporal discrepancy makes it difficult to conclude that NADPH/NADH gradient (E6) originates from the glucose uptake gradient (E14). Similarly, it is difficult to link a causal relationship of tonotopic differences between surface area and glucose uptake because while glucose uptake is not significantly different until E14 (Figure 4), the surface area can be changed by blocking glycolysis at E6-E8 (Figure 5). To argue that metabolic gradients play a role in tonotopic patterning, the tonotopic gradient of NADPH/NADH, its regulators, and glucose uptake should be clearly correlated spatially as well as temporally in the developing BP.

The temporal gradients will not necessarily match as they report different steps of the glucose catabolism pathway. 2NBDG reports uptake of glucose into the cell via GLUT transporters. NADPH/NADH reports differences in the path of glucose catabolism once it has been phosphorylated by HK. Because 2-NBDG cannot by phosphorylated or metabolised by HK, it does not reflect the different paths by which glucose is metabolised. We hope this clarifies the reviewer’s concern and have re-written this in the Results section: see section beginning Line193 – “Live imaging of mitochondrial metabolism and glucose uptake along the tonotopic axis of the developing BP…”.

…it difficult to conclude that NADPH/NADH gradient (E6) originates from the glucose uptake gradient…

We thank the reviewer for raising this question and have made attempts to clarify this in the text. The gradient in NADPH reported by tbound is not a reflection of glucose uptake but of glucose flux into the PPP or NADPH production in mitochondria. We also acknowledge that tbound can of course report a mixture of these two metabolic processes. We have now included a schematic (see Figure 1) illustrating the different metabolic branches and pathways of glucose catabolism.

4) Changes in Bmp7 and Chdl1 gradients by 2-DOG treatment (Figure 8) provide a molecular mechanism by which glucose metabolism regulates the tonotopic patterning of BP. However, 2-DOG has been shown to influence several cellular processes other than glycolysis, such as angiogenesis and autophagy. Therefore, it is crucial to confirm that the changes in Bmp7 and Chdl1 gradients by 2-DOG are due to inhibition of glucose metabolism but not due to other cellular effects.

In attempt to address this issue we chose to use a combination of different inhibitors. As inhibitors of glycolysis (2-DOG, SAM) and of the PPP (6-AN) elicited the same reduction in luminal SA and nuclear area, we conclude that it unlikely for all inhibitors to elicit the same off target effects on cell morphology.

This will be confirmed if SAM and 6-AN, but not YZ9 and UK5099, which induced surface area changes similar to 2-DOG, also cause similar changes in Bmp7 and Chdl1 gradients.

In situ for Bmp7 following UK5099 treatment has been included. We did not see any notable change in Bmp7 in whole mount in situ after treatment with UK5099 for 7 days in vitro (see Figure 9 Supplement 1). We have also included new data showing the effects of 2-DOG treatment in the presence of Chordin-like 1 (Figure 10).

1) Specify the model system in the abstract since birds and mammals appear to have different mechanisms for tonotopic patterning.

This has been specified and the definition/abbreviation added to abstract.

2) Figure 1I., τbound values were consistently higher in the proximal than distal regions from E6 to E16, although statistical analysis indicates significant differences at E6 and E14 but not at E9 and E16. Non-significance at E9 and E16 may be due to the small number of n, which made statistical power not strong enough to draw statistical significance. To give the readers a better idea of how variable τbound values are during development, it is recommended that the graphs on the left are presented as a boxplot with individual data points. Also, provide error bars and statistical analysis for the graphs on the right.

Additional replicates have been added and all data are now significant at E6, E9 and E14 but not E16. However, by E16 HCs are fully differentiated so it is assumed that tonotopic patterning would be established at this stage.

3) Figure 2C and D. Similar to the point above, it is recommended to present the graphs as a box plot with individual data points. Does "n = 10 biological replicates" mean 10 different BP samples? If so, indicate how many hair cells and supporting cells per BP sample were used to calculate τbound values?

Cell numbers are now indicated, and box plots have been used throughout to show individual data points.

4) Line 323. There are several instances where the authors refer to "other systems". It will be better to specify the system the authors refer to so that the readers get an idea of how the "other systems" are related to the auditory system. e.g., "studies in other systems" in lines 458, 505, and 685.

The text has been significantly re-written. This statement is no longer included in the article.

5) Figure 8. Why are the surface areas of control BPs so different between B (~40 μm2 in proximal) and C (~20 μm2 in proximal)?

In Figure 9B measured HC liminal SAs in 2500 µm^2^ areas. Figure 9C measure HC luminal SAs in 10000 µm^2^ SAs.

6) Figure 8 Supplement 1. It is difficult to conclude based on the images whether 2-DOG treatment affects proliferation in the developing BP. Counting EdU+ cells will help to conclude whether or not cell proliferation is changed by inhibiting glucose metabolism.

Additional EdU experiments have been added and cell numbers have been quantified in proximal and distal regions. See Figure 6 Supplement 2.

7) Figure 9. The results from the mouse organ of Corti are interesting but do not appear to be relevant to the rest of the manuscript, since differences in τbound values were compared only between HCs and SCs, but not along the tonotopic axis. It is recommended to either present the τbound values between the proximal and distal regions of the organ of Corti or save them for further study.

This figure has now been removed from the paper.

8) Line 853. While the Shh gradient is present along the auditory organ in both birds and mammals, the Bmp7 gradient is present only in birds but not in mammals (Son et al., 2014, PNAS).

We thank the reviewer for drawing attention to this error. Reference of these data in the text has been corrected. See line 844.

9) It is clear that inhibition of glucose metabolism by 2-DOG, SAM, and 6-AN induces the proximal BPs to become more like distal BPs, such as decreased surface area and increased Bmp7 expression in the proximal region. Is it possible to conduct the opposite experiment, i.e., activation of glucose metabolism, which may cause distal BP to become like proximal BP?

There are currently no pharmacological activators of glycolysis of PPP flux. The only way to activate glucose catabolism in different branches would be through the over expression or knock-down of specific rate-limiting enzymes using in ovo electroporation. Whilst important experiments to address, we believe these go beyond the scope of the current study.

Reviewer #3 (Recommendations for the authors):In this manuscript, Blacker et al. examine cellular metabolism primarily in the chick auditory organ, the basilar papilla (BP). They use FLIM to visualize NAD(P)H in live tissue to study changes in glucose metabolism within the auditory epithelium over the course of development. They show that there are differences in metabolism between HCs and SCs. They also show that there are spatial differences in metabolism along the length of the BP at specific developmental time points. The authors bolster evidence for this metabolic gradient using existing RNAseq datasets that show differential expression of metabolic genes along the proximal-to-distal axis of the BP. They then use pharmacology to manipulate glucose metabolism (glycolysis, pentose phosphate pathway, and the TCA cycle). The authors show that altering glucose metabolism early in glycolysis or at the level of the pentose phosphate pathway results in a loss of tonotopy in the BP. In contrast, inhibiting glucose metabolism at the level of the TCA cycle impaired HC differentiation. Further, they show that inhibiting glycolysis alters morphogen gradients that are required for the emergence of tonotopy within the developing BP. Overall this work suggests that specific metabolic pathways may drive developmental changes in morphogen gradients and – perhaps – contribute to HC differentiation. This is an exciting advance for the hearing field, especially with respect to potential regenerative therapies.Strengths of the paper:1. The use of FLIM to study NAD(P)H fluorescence in living HCs and SCs across the auditory epithelium during the course of development.2. The powerful use of pharmacology to probe the role of glucose metabolism in tonotopy in long-term cultures of BP explants.3. Using multiple approaches for the study: live NAD(P)H imaging, RNAseq data, pharmacology, vital dyes, and immunohistochemistry.4. Linking distinct metabolic pathways to developmental events is a novel and exciting area of research.Weaknesses of paper:1. The majority of the manuscript uses t-tests for comparisons. In many cases, a 2-way ANOVA is needed for comparisons. It is not clear that all the findings will hold up to more rigorous statistical analysis.2. Substantial work is needed to enhance the clarity of the manuscript. There is a limited introduction to the different facets of glucose metabolism highlighted in this paper. Metabolism is a complex topic that a broad audience will not be intimately familiar with. In addition, the paper is rich with data. The authors discuss differences in metabolism along 3 variables (space, time, and cell type). Occasionally it is unclear what type of difference is being discussed and the trend of the difference.3. Manipulations inhibiting glycolysis and oxidative phosphorylation over extended periods of time could have many secondary effects on development related to a lack of energy supply.Overall, this is a very interesting line of research. Currently, the manuscript needs revision prior to publication.1. Statistics. The majority of the manuscript uses t-tests, which are not appropriate when making multiple comparisons (ie: proximal vs. distal over multiple stages of development). Currently, it is not clear that all the findings will hold up to more rigorous statistical analysis. For example, Figure 1I, 1J, 2C, 2D, 4D, 4F, 4G, 4I, 4J, all of Figure 4 Supp. 1, Figure 4 Supp. 2B, 5C, Figure 5 Supp. 1C, 6D, Figure 6 Supp. 1B, 7C, 8B, 8C, 9E, and 9F should be analyzed using two-way ANOVA. It is not statistically sound to pick only specific comparisons to make.

All statistical analysis has been re-done using 2-way ANOVA testing. We refer the reviewer to the data now included in table 1.

Additionally, in some cases, there are stars above bars with no indication of what condition is being compared to (ie. Figure 9E). Figure 3A lacks statistical analysis altogether. Finally, there are several cases where the authors say that there is a "significant" difference but do not show that it is statistically significant (for example, the caption of Figure 4I says there is a significant decrease between E9 and E14, but there is no indication of statistical significance in the figure). Differences that are not statistically significant can be discussed as trends, but should not be called significant. Similarly, differences that are statistically significant should not be discussed as "no difference" (ex. Figure 4E/F caption but Figure 4F shows a statistically significant difference; Figure 4 Supp. 2).

All concerns regarding significance and correct indication of this on figures have been addressed.

2. Enhance the clarity of the manuscript. The manuscript is quite well written but currently, there is limited text in the introduction outlining the relevant parts of glucose metabolism highlighted in this paper. Or how NADPH and NADH feed into these pathways. In addition, the authors discuss differences in metabolism along 3 variables (space, time, and cell type), and occasionally it is not clear what type of difference is being discussed – what is increased or decreased and where. Schematics like those in 2E are helpful to the reader.

Schematic indicating the different metabolic pathways and path of glucose catabolism has been added (see Figure 1).

3. Manipulations inhibiting glycolysis and oxidative phosphorylation over extended periods of time could have many secondary effects on development. The authors should lighten claims of causation. At a minimum, there should be a discussion of possible secondary effects and the need for more targeted manipulations to draw a convincing causative link between cellular metabolism and the development of tonotopy within the BP.

This section has now been significantly revised and schematics have been added to show cell-specific expression of metabolic enzymes and the metabolic processes they regulate.

4. The authors detect a proximal-to-distal gradient in tbound at E6 that "was still present at E14" and discuss this as a gradient "along the tonotopic axis between E6 and E14". However, their figure does not show a significant difference between the proximal and distal regions at E9. This could be related to a lack of statistical power (the n is lower for E9), but could alternatively represent a real fluctuation in the metabolic state over time. The authors should include a discussion of their E9 data, rather than glossing over it.

This concern has now been addressed in the discussion and the use of multiple inhibitors to target the same pathways was how we chose to address the concerns of off-target effects.

5. Although it may be beyond the scope of this paper, it would be nice to come full circle back to NAD(P)H imaging. Do the alteration to the PPP pathway lead to a loss of the NADPH/NADH gradient along the proximal-distal axis in the developing BP?

Following analysis using 2-way ANOVA the gradient in NADPH is significant at all stages from E6-E14.

6. The authors make several claims regarding differences in TMRM. They should also consider that this cationic vital dye may enter HCs via mechanotransduction channels. This form of entry may explain why labeling is stronger in older HCs compared to SCs or younger HCs.

This is important point and something we plan to address with future experiments. To do the FLIM imaging in acute explants we used morphological landmarks i.e otoconia of the utricle, saccule and Lagena at early stages and at E9 and E14 it is easy to differentiate between base and apex using the shape of the epithelium. This becomes significantly more difficult after 7 days in culture as the shape of the epithelium is slightly distorted. In the absence of cell-specific markers such as Phalloidin and calbindin. These experiments would require in ovo manipulation of metabolism.

[Editors’ note: what follows is the authors’ response to the second round of review.]

The manuscript has been improved but there are some remaining issues that need to be addressed, as outlined below:Essential revisions:While all three reviewers felt that the revised manuscript is a substantial improvement there are still some outstanding issues:1) The Chordin rescue experiment requires additional controls and clarification.

We fully agree with the reviewer regarding the necessity of this data and have thus included dose response experiments for Chordin like-1 in addition to media only controls and Chordin-like 1 treatments in the same experimental paradigm. Please see new data and analysis included in the revised Figure 10. All doses of Chordin like-1 are now clearly stated in Figure 11, in the main text (see line 348) and in the Methods section of the article (line 494).

We would also like to lessen the emphasis on any claim that metabolism is acting as a master regulator of HC tonotopy and instead highlight that a causal interaction between cytosolic glucose metabolism and Bmp7/Chdl-1 signalling is necessary for specifying HC size and regulating patterning of the mosaic at the epithelial surface (see revised paragraph beginning line 328).

2) Additional inconsistencies and quantification were requested by all three reviewers.

New data and additional quantification included in the manuscript as requested by the reviewers.

Additional quantification of hair cell density and nuclear area have been included for 2DOG, SAM, 6AN, YZ9 and shikonin.Additional experiments have been included investigating a role for PKM2 in tonotopic patterning during development (see Figure 8).As highlighted by all three reviewers, the previous expression data for IDH3A and LDHA did not fit logically within the study and caused confusion about which metabolic pathway was being probed. The data for IDH3A and LDHA are therefore no longer included in the current manuscript and the text has been revised accordingly.Quantification of EdU positive cells has now been included (Figure 6 Supplement 2).Cell density analysis has been included for Chdl1+2DOG and Chdl1 (0.4 g/mL) alone (new Figure 11 Supplement 1).Dose response data have been added for Chdl1 as requested by the reviewers (Figure 11)Further statistical tests have been performed on proximal vs proximal treated and distal vs distal treated explants (see revised Figure 6).Further experiments have been conducted to investigate the pH gradient along the BP at more mature developmental stages. (see new Figure 5 Supplement 3).Additional RNA scope analysis has been included for both *BMP7* and *PKM2* at E8, showing the tonotopic expression gradients for both genes (see new Figure 3 Supplement 4 and Figure 4 Supplement 1). New data showing the difference in PKM2 expression between proximal and distal regions has also been included at E10 (see Figure 4 Supplement 2) and E12 (see Figure 4 Supplement 3).Negative RNA scope probe controls have been included for *BMP7*, *PKM2* and *SOX2* (see Figure 3 Supplement 5).Additional in situ data have been included, showing BMP7 and CHDL1 expression following treatment with 6AN (see Figure 10 Supplement 1).Data showing explants treated for 72h with UK5099 has been added as a Supplement to Figure 9 (new supplementary Fig?). As recommended by the reviewer ??, these data aim to support the claim that blocking mitochondrial OXPHOS causes significant a developmental delay (see Figure 9 Supplement 1).

3) Comparing treatments of SAM or 6-AN to 2-DOG in the ability to affect Bmp7 signal, for example.

Additional in situ experiments for *BMP7* and *CHDL-1* have been conducted with 6-AN (Figure 11 Supplement 1).

Reviewer #1 (Recommendations for the authors):This revised manuscript is much improved and streamlined. The authors provided additional results showing that Chordin like-1, which is normally expressed in the proximal region of the BP could rescue the distalizing effect of 2-DOG in the proximal region, thus strengthening the link between metabolism and tonotopy. However, in this type of rescue experiment, 2-DOG and Chordin like-1 (Chdl-1) treatments should be included as controls within the same experiment, rather than using controls with no treatment alone. While readers could extrapolate 2-DOG effects from other figures, there was no description of Chdl-1 in dosage, condition, or results in the revised manuscript. Furthermore, the statement in lines 800-802 and results in Supplementary Table 1 seem to suggest that Chdl-1 treatment alone does not affect the tonotopic gradient by itself. Is that true? If so, this result is not consistent with previous studies.

Chdl-1 dose response experiments are now included in Figure 11 with doses clearly indicated.

Reference to the dose used for treatments (0.4 g/mL) is now stated in the main text of the Results (lines 348 and 494) and in the Methods. Treatments with Chdl-1 alone show disruption of the normal tonotopic gradient. These data are included in Figure 11 and highlighted additionally in the statistical data presented in Supplementary Tables 1 and 2.

Editorial suggestions to help improve the manuscript:1) Line 28, what does flattening of hair cell morphology mean? Do you mean shorter hair bundles in the proximal versus the distal BP? If so, the images in this study are not clear enough for naïve readers to draw that conclusion.

This statement has been changed in the Abstract and in the main text to read:

“…abolishes tonotopic patterning and normalises the graded differences in hair cell morphology along the BP…” (Line 26).

The phrasing in Line 214 has also been changed for clarification.

2) Line 54, what does inner sense organs means? Are you referring to the inner ear or tissues deep in the body?

We have changed this to ‘developing inner ear epithelia’ (line 53).

Labels have been added to indicate the different cellular compartments.Numbering has been removed for clarity.Font size has been increased for all pathways.Mitochondrial membrane has been more clearly labelled.Site of shikonin (PKM2 inhibitor) activity added to schematic.4) Lines 293 and 295, a bit odd to split PKM2 results into two separate paragraphs.

This section of the manuscript has been significantly re-written. The paragraphs for PKM2 are now combined and additional data has been included (see section beginning line 160).

We also refer the reviewer to the new data for PKM2 expression included in Figure 4 Supplements 1, 2 and 3.

5) Figure 2 a) Consider combining panels D and K, and expand B and C. Hard to read the label in C, b) I am still not clear why αbound level (dotted purple line) is in a steady state based on the purple and green curve and there is no explanation of αbound in the text, c) Panels E-H could be slightly bigger, d) move Scale bars=50um to the end of description for (E-H).

We thank the reviewer for their important comments. We have addressed all recommended edits and hope that the revised Figure 2 is now much clearer.

6) Line 205, probably better just to state τbound gradient not related to glucose uptake?

This statement has been reworded as follows:

The gradient in NADPH/NADH reported by τ_bound_ (Figure 2 D-I) does not arise from variations in the balance between cytosolic and mitochondrial ATP production, as often observed in development but from differences in the route of cytosolic glucose processing utilised. A new schematic has also been included for clarification of what τ_bound_ is showing (see Figure 2 Supplement 1).

7) Line 236, the sentence is a bit awkward.

Thank you and this sentence has been reworded.

8) Line 292, define PKM2 when it first appeared in the text.

Pyruvate kinase M2 defined in main text at first mention.

9) Line 339, Figure 3 or Figure 5?

Corrected to read Figure?

10) Line 342, Figure without a number.

This Figure/text has been removed.

11) Lines 340-343, State in the text lower pHrodo Red signal meant higher pH.

Thank you for this comment and we have now clearly stated the in both the legend for Figure 5 and in the main text.

12) No quantification of Edu-labeled cells in Figure 6 suppl 2.

We apologize for not including quantification, which has now been added.

13) Line 484, does SAM inhibit both glycolysis and PPP pathways similar to 2-DOG?

Beginning line 243 “To further confirm a role for cytosolic glucose metabolism in establishing HC positional identity, we employed a second method of modulating the pathway independently of HK inhibition. Cytosolic glycolysis can be blocked indirectly by increasing cytosolic levels of the metabolite s-adenosyl methionine (SAM).”

In response to the reviewer, SAM does not act in the same way as 2DOG to block cytosolic glycolysis. 2-DOG inhibits at the level of hexokinase, the first enzyme in the man glycolysis pathway and SAM affects glycolysis indirectly by altering the activity of metabolic enzymes in the pathway

(Pascale et a; 2019) doi:10.3390/medicina55060296

(Yu et al., 2019) doi:10.1016/j.molcel.2019.06.039. Epub 2019 Aug 13

We hope this is has been sufficiently clarified in the revised main text. See paragraph beginning line 242.

14) Typo in the last sentence of the legend for Figure 7 supplement 1.

Corrected.

15) Incomplete last sentence found in the legend for Figure 8.

Corrected.

16) Figure 9. The images in E are better in this revision. However, there is some inconsistency between the text and the figure legend. In the text, explants were established at E6.5 and incubated for 72 hours. In the legend, it was stated the in vitro incubation was for 7 days.17) Figure 9. It may be better to put the number and statistics in the panels that they belong rather than at the end of the figure. Sample numbers need to be provided for panel E as well.

We thank the reviewer for identifying the error in the Figure legend. The legend for Figure 9 (now Figure 10) has been corrected to read:

“…treated with 2-DOG +NaP from E6.5 for 72 hours in vitro, as stated in the main text of the article.”

Legend order has been rearranged as suggested by the reviewer. Samples numbers have been added to panel E.

18) Supplemental Table 1. Are all p values in the left-hand columns controls? Confused by the statement in line 799, "control or inhibitor treatment groups". Or it should be controlled for inhibitor treatment groups? If so, the p values in the right-hand column are results from individual treatments. Then, Chdl-1 treatment did not affect the tonotopic differences observed in controls?

We refer the reviewer to the amended version of Supplementary Table 1. For clarification, data have been divided into statistical outcomes for luminal surface area and nuclear area. Please see new Supplementary tables 1 and 2.

Reviewer #2 (Recommendations for the authors):Figure 7 supplementary 2.Based on the statistical analysis, the authors concluded that YZ9 treatment had no effect on HC morphology along the tonotopic axis. However, the graph in (B) shows that the luminal area of hair cells was dramatically reduced in the proximal but not in the distal region by YZ9 treatment, resulting in no significant difference between the proximal and distal regions. This suggests a loss of tonotopic identity of HC morphology by YZ9 treatment, similar to 2-DOG, SAM, or 6-AN. To support the authors' argument, it is important to confirm that YZ9 treatment did not induce tonotopic changes in HC morphology. Double-checking with other tonotopic parameters, such as HC nuclear area and cell number, will help to clarify this issue.

We thank the reviewer for their comments and recommendations to clarify this issue. Additional quantification of HC number and nuclear area have been included for all inhibitor treatments and statistical data added to Supplementary Tables 1 and 2.

Figure 9.(E) The Bmp7 expression domain in the control explant is peculiar in that it has a lining around the distal region, unlike the control image in Figure 9 supplement 1. Figure 3 supplement 4 shows that Bmp7 is normally expressed in the sensory domain.Bmp7 in situ signals are generally stronger in 2-DOG-treated BP than in control, but there still seems to be a proximal-distal gradient with a generally stronger in situ signal. qRT-PCR with each half of the BPs, as done in their previous publication, will clearly show if the gradient is disrupted.

We thank the reviewer for raising this concern, and fully agree that from the in situ data alone it is challenging to make any assumption regarding the change in either gradient.

We would like to clarify that we are not proposing that glucose catabolism sets up either of the morphogen gradients. We believe the Bmp7 gradient is set up, as shown by Son et al. (2015) by the activity of Shh from the ventral midline structures.

The new in situ data included in Figure 10 Supplement 1 highlight the complicated regulation of *BMP7* and *CHDL1-1* expression by different metabolic pathways.

2DOG causes an increase in *BMP7* and a decrease in *CHDL-1* along the entire organ but, as highlighted by the reviewer the gradients of both morphogens appear maintained. We therefore feel that qRT-PCR analysis is unlikely to reveal different results regarding either expression gradient. 6AN reduces *CHDL-1* in the proximal but not distal BP region and causes a subtle expansion of BMP7 towards the proximal end of the BP (see new Figure 10 Supplement 1). The reduced *CHDL-1* combined with increased *BMP7* following treatment with either 2DOG or 6AN would however be sufficient to induce more distal-like HC morphologies in the proximal region of the tissue (please see Figure 10 and Figure 10 Supplement 1).

We propose that the concentration of *BMP7* and *CHDL1* seen a HC at a given point along the BP, coupled with a unique metabolic phenotype regulates epithelial patterning and HC shape (HC nuclear area, LSA and density) along the developing BP. Without having performed a detailed physiological analysis, we would be reluctant to make any firm conclusion regarding the ‘HC fate’ or frequency tuning that accompanies the observed morphological changes. Understanding the precise mechanism linking cell shape with physiological function along the developing BP requires further investigation, which we feel is beyond the scope of the current study.

Figure 9 supplement 1.The previous review questioned whether the changes in tonotopic identity caused by 2-DOG were due to the inhibition of glucose metabolism rather than other cellular effects. This question was crucial to support the authors' conclusion, as 2-DOG has been shown to affect several cellular processes other than glycolysis. The authors showed that blocking mitochondrial metabolism did not alter the Bmp7 gradient, but this doesn't solve the problem of the specificity of 2-DOG.

We agree with the reviewer that when using pharmacological inhibitors there are likely to be offtarget effects. Given the conserved morphological changes elicited across multiple inhibitors (see Supplementary Tables 1 and 2), we are confident however that the observed effects on cell morphology arise due to perturbation in glucose flux and not from off-target effects of the drug.

We agree that this is an important point to address and therefore offer the following response to address the concerns raised by the reviewer.

2-DOG is a widely accepted inhibitor of hexokinase and thus glycolysis, so much so that it is the biochemical basis for PET scanning in cancer. Off-target effects are not widely reported. We offer the following citations for 2DOG in hope to satisfy any concerns raised by the reviewer:2-DOG can stimulate IGF1R signalling (https://doi.org/10.1074/jbc.M109.005280), thereby activating Akt by inducing its phosphorylation (https://doi.org/10.1158/1535-7163.MCT-070559). However, Akt would be unable to activate glycolysis under these conditions as hexokinase would remain inhibited.2-DOG can also activate autophagy pathways (https://doi.org/10.3390/ijms21010234) but this further is downstream of glycolysis, due to the resultant reduced ATP/AMP ratio.It can inhibit N-glycosylation, leading to ER stress (https://doi.org/10.3390/ijms21010234). However, this results from the structural similarities between D-glucose and D-mannose, so if an alternative way to inhibit cytosolic glucose metabolism has the same effect, this offtarget effect (i.e. SAM, 6AN and shikonin) cannot be responsible for our observations.

It is worth investigating whether SAM and 6-AN, which induce luminal area changes similar to 2-DOG, also disrupt the same molecular pathways, such as the Bmp7 and Chld1 gradients.

New in situ data have been included for treatments with 6AN and 6AN + shikonin (see Figure 11 Supplement 1).

Reviewer #3 (Recommendations for the authors):This manuscript is substantially improved compared to the previous version. Thank you for all the hard work you put into responding to reviewer concerns. A revised manuscript would be welcomed by this reviewer.– The RNAseq and microarray analyses are good for hypothesis generation, but the differential expression the authors detect often does not match the immunohistochemistry and RNA scope data they provide. It is especially confusing to say that there is no difference in LDHA and IDH3A protein expression across the BP, then suggest that differential RNA expression of these genes could drive the metabolic gradient the authors observe.

We agree fully with the reviewer that the original data included in Figure 4 did not fit logically with the proposed hypotheses of the study. We have since removed these data, as the focus of the study is about differences in cytosolic glucose flux along the organ and not metabolic switching. We have included a new schematic (Figure 2 Supplement 1) to highlight to metabolic differences reported by FLIM analysis. As IDH3A and LDHA regulate switching between OXPHOs and glycolysis, we have revised the current version of the manuscript accordingly and focused more on the role and expression of PKM2, an enzyme that regulates both glucose flux into the PPP, and the NADPH/NADH ratio. We hope that this resolves the previous confusion.

Some of the variability in the data may be related to differences in animal age across experiments, but this variability makes it a bit challenging to interpret the data. In addition, not all the genes listed in Figure 3 Supp. 4A seems to reach statistical significance. The authors should consider framing Figure 3 Supp. 4A as a way to generate hypotheses that they then test with additional experiments, rather than claiming that they identified a whole set of differentially expressed genes when in fact they present contradictory data within the same manuscript. They should also clarify how they decided which metabolic RNAs to examine further.

We fully agree with the reviewer’s suggestion regarding the RNAseq and RNA scope data and have therefore conducted additional analysis and experiments investigating the role of the gate keeper enzyme PKM2 (Figure 4 Supplements 1-3, Figure 8, Figure 11 Supplement 1). PKM2 was chosen for further analysis, as it regulates glucose flux into the PPP or towards pyruvate and the NADPH/NADH ratio. This has also been clarified in the main text (see lines 157, 160 and 175).

– Although the authors used BMP7 as a positive control for RNA scope, there does not appear to be a proximal-to-distal difference in BMP7 fluorescence.

Thank you for this important observation. To address this point, new data showing serial crosssection of the BP at E8 has been included showing the distal-to-proximal gradient in Bmp7 (Figure 3 Supplement 4).

In addition, the authors do not provide any negative controls to show a lack of fluorescence in the absence of gene-specific RNA scope probes. Without successful controls, the experimental data is hard to interpret. The authors should consider providing better controls.

We concur with the reviewer and have now included RNA scope controls for all probes used (Figure 3 Supplement 5).

– Line 297-298: "By controlling a metabolic feedback loop that regulates the switch between glycolysis and OXPHOS…" The authors previously showed data suggesting that differences in the NAD(P)H/NADH ratio are not due to changes in mitochondrial OXPHOS. Here, though, they suggest that PKM2 controls the amount of OXPHOS and is differentially expressed across the BP. How can we reconcile these two points?

In the current study, we propose that the dimeric form of PKM2, regulated by pH might control glucose flux into the PPP or towards pyruvate. Given the low pH, high NADPH/NADH ratio and expression of PKM2, our data suggest activity consistent with glucose flux into the PPP rather than pyruvate production. This would agree with the lack of a TMRM gradient and thus mitochondrial OXPHOS along the BP.

Please see the revised paragraph beginning line 188.

– Figure 4, 5: How was the quantification in 4B-D and 5B performed? Are fluorescence values provided per HC/per SC? As HC size varies across the proximal-distal axis of the BP, it is not fair to quantify raw total fluorescence. The fluorescence should be normalized by the area occupied by each cell type.

IDH3A and LDHA immunofluorescence data are no longer included in the current version of the manuscript. Additional information on how the analysis was performed has however been included with Figure 5 (Figure 5 Supplements 1 and 2).

– Figure 5: Could proximal-to-distal differences in pHrodo Red fluorescence result from the gradient of HC differentiation (i.e. more mature distal HCs, less mature proximal HCs) across the tonotopic axis at E9? In other words, do young HCs have a higher pHi that decreases as hair cells mature? This could explain the proximal-to-distal differences shown in Figure 5.

We have included additional data investigating the pH gradient at E14 (see Figure 5 Supplement 3).

At later stages, as predicted by the reviewer, we do indeed find that the pH gradient is reversed. However, we are confident this does not affect specification of positional identity in nascent HCs, as tonotopic patterning is established between E6-E7 (Mann at al., 2014).

– Figure 6 Supp. 2 – To claim that there is no increased proliferation in response to 2-DOG, EdU+ cells should be quantified in control vs. 2-DOG cultures. It seems that the authors have only provided general cell counts (in C) and not EdU+ cell counts.

We fully agree with the reviewer and have now included new EdU quantification for this experiment (see Figure 6 Supplement 2).

– Do the authors have any evidence that the various drug treatments are not simply blocking cell growth and differentiation? As mentioned by another reviewer, the distal-like phenotype could be a "default"; perhaps proximal HCs simply aren't maturing. It would be nice to show examples of cultures at 1DIV vs. 7 DIV to demonstrate how much cells grow during this time and how treated cultures at 7 DIV compare to immature 1 DIV cultures. (SAM treatment especially looks like it might impact hair bundle morphology in the proximal BP.)

We thank the reviewer for raising this important point and, to address the concern, have included additional data showing the effect of UK5099 during short-term culture. We are confident that blocking mitochondrial OXPHOS causes developmental delay, but that perturbing glucose flux alters normal patterning. As indicated in Figure 9 Supplement 1, the bundle morphology at earlier stages in cultures (E9.5) is immature compared to cultures left to develop for the full 7 days in culture, even in the presence of cytosolic glucose flux inhibitors.

– Figures 7 Supp. 2 and Figure 8 continue to rely on a single morphological measure of HC morphology as a readout of tonotopy. The authors should add additional quantification (cell density, nuclear area…), as has been provided for other manipulations.

As recommended by the reviewer, additional quantification of cell density and nuclear area have been included for all inhibitor treatments. Please see revised Figure 6, Figure 6 Supplement 3, Figure 7, Figure 7 Supplement 2, Figure 8 and Figure 11.

– Figure 9 and Figure 9 Supp. 1: In the absence of quantification of the in situ hybridization data, please state how many examples show similar Bmp7 and Chdl1 gradients.

Information regarding the numbers of biological replicates have now been clearly stated in the legends for revised Figure 10 and Figure 10 Supplements 1 and 2.

– To bolster the claim that 2-DOG treatment is specifically disrupting tonotopy and not just the differentiation of proximal HCs (as pointed out by Reviewer 1), the authors should quantify HC density in Figure 10. If HC density is also restored by providing Chdl-1 in the presence of 2-DOG, it will bolster the hypothesis that altered glucose metabolism has a specific effect on HC positional identity.

We fully agree with the reviewer and have therefore included quantification of HC density in control, Chdl1+2DOG and Chdl1 0.4 g/mL treated explants. Please see Figure 11 Supplement 1.

[Editors’ note: what follows is the authors’ response to the third round of review.]

The manuscript has been improved but there are some remaining editorial issues that need to be addressed, as outlined below:Reviewer #1:The results of the revised manuscript are coming together nicely. To appropriately demonstrate the rescue effect of Chd1 on 2DOG requires the control, 2-DOG, and 2-DOG+Chd1 be all conducted within the same experiment. Although such an experiment was not provided, the authors have made a good faith effort in demonstrating a dose-response effect of chordin in reducing the differential differences between proximal and distal BP, and the revision is acceptable. Additionally, I have some editorial suggestions. These are merely suggestions, and the acceptance of the manuscript is not dependent on making these changes.1) The experimental evidence suggests that the tonotopic morphogens are regulated by glucose metabolism. I don't see evidence showing that these morphogens regulate metabolism in return. Therefore, I question the choice of using the phrase "Cross talk" in the title.

We agree with the reviewer on this point and have therefore re-worded the title as follows: Causal gradients in glucose metabolism and morphogen signalling specify tonotopic identity in developing hair cells.

2) The abstract, though clear, could be more concise and informative. I feel like it is important to summarize this 32-figure manuscript well!

We thank the reviewer for this suggestion and agree regarding the importance of a concise, clear summary. We have therefore reworded the abstract and hope this more clearly outlines the take home findings of the study.

3) Include PKm2 in the schematic diagram of Figure 1.

Pkm2 (pyruvate kinase) has been added to the schematic in Figure 1.

4) The first part of the Discussion reads more like an introduction.

We appreciate this comment and have re-worded the opening paragraph of the discussion.

Reviewer #2:The revised manuscript is much improved with additional experimental data that strongly support the authors' conclusions. I appreciate the authors' efforts to make the manuscript in much better shape.

We very much thank the reviewer for their favourable comments in response to the revised version of this article.

I have just one question. From the diagrams in Figure 7-supplement 2B and Figure 8B, it appears that PFK and PKM2 act in the same glycolysis pathway. However, inhibition of either resulted in different phenotypes. While inhibition of PFK by YZ9 did not affect tonotopic patterning, inhibition of PKM2 by shikonin disrupted tonotopic patterning, albeit at the epithelial level. Why did two inhibitors of the same pathway have different effects on tonotopic patterning?

This is an important point, and we thank the reviewer for drawing attention to it. The following explanation has been added to the main text for clarification – line 285. We would like to clarify that although YZ9 (Pfk) and shikonin (Pkm2) both inhibit parts of glycolysis, noting the two enzymes regulate different stages of the pathway with respect to glucose flux and the re-entry of PPP products. Inhibition of Pkm2 would cause build-up of both G3p and F6p, whereas inhibition of PFK would only lead to accumulation of F6P. From this it can be assumed that Pkm2 inhibition would have a greater impact on the activity of the PPP and therefore, given the gradient in NADPH, on tonotopic patterning.

Reviewer #3:The manuscript is greatly improved and nearly ready for publication.There are a few typos and points to clarify that I would recommend to the authors. These recommendations should not require additional experiments.List of recommendations for clarity:Figure 2 Supplement 1 – this Figure is really helpful and we would recommend making it part of a main figure (perhaps you can condense Figure 2G, as these two plots show the same data, in favor of this helpful schematic). The clarity of the FLIM section is massively improved since the first version of the paper. The one link that is still hard to make is how the values of Tbound inform flux through the PPP vs. glycolysis. This figure beautifully illustrates exactly that. Also, consider adding PPP vs. glycolysis to the gradient in the Figure 2H schematic, and/or moving the sentence in line 109 to line 101 in the main text. Although all of this knowledge is obvious to those who think about metabolism all the time – these concepts can be confusing for someone not in the field.

We thank the reviewer for these helpful suggestions and agree that these changes make Figure 2 significantly clearer. Please see revised Figure 2, which now contains the recommended edits. To make the description of our data in this section of the results clearer, we have moved the sentence from line 109 to line 101 as recommended by the reviewer.

We would recommend adding Pkm2 to Figure 1, as there is an entire figure focused on Pkm2.

We have now added pyruvate kinase (Pkm2) to the schematic in Figure 1.

Figure 3 Supp. 3E. Are you detecting statistically significant proximal-to-distal changes in Bmp7 here? It is not indicated in the quantification (no stars or "ns"). If the positive control is not showing a statistically significant change, it is hard to interpret the other data.

Although the distal-to-proximal gradient of Bmp7 was consistent in all 5 biological replicates, the high variation across samples meant that significance was not reached in the data. To address this issue, we therefore provided additional confirmation of the distal-to-proximal gradient using serial sections along the entire proximal-to-distal axis. The other mRNAs (Got2 and Ldhb) shown in Figure 3 supplement 3 were not significantly different along the BP and were therefore not pursued further in relation to the NADPH gradient along the BP.